# Reduced calcium levels and accumulation of abnormal insulin granules in stem cell models of HNF1A deficiency

Bryan J. González [1,2], Haoquan Zhao[3], Jacqueline Niu[4], Damian J. Williams[5], Jaeyop Lee[3], Chris N. Goulbourne[6], Yuan Xing[7], Yong Wang[7], Jose Oberholzer[7], Maria H. Blumenkrantz [1], Xiaojuan Chen[8], Charles A. LeDuc [1], Wendy K. Chung [1], Henry M. Colecraft [4], Jesper Gromada[9,10], Yufeng Shen [3], Robin S. Goland[1], Rudolph L. Leibel[1] & Dieter Egli [1✉]

Mutations in *HNF1A* cause Maturity Onset Diabetes of the Young (HNF1A-MODY). To understand mechanisms of β-cell dysfunction, we generated stem cell-derived pancreatic endocrine cells with hypomorphic mutations in *HNF1*A. *HNF1A*-deficient β-cells display impaired basal and glucose stimulated-insulin secretion, reduced intracellular calcium levels in association with a reduction in *CACNA1A* expression, and accumulation of abnormal insulin granules in association with *SYT13* down-regulation. Knockout of *CACNA1A* and *SYT13* reproduce the relevant phenotypes. In *HNF1A* deficient β-cells, glibenclamide, a sulfonylurea drug used in the treatment of HNF1A-MODY patients, increases intracellular calcium, and restores insulin secretion. While insulin secretion defects are constitutive in β-cells null for *HNF1A*, β-cells heterozygous for hypomorphic *HNF1A* (R200Q) mutations lose the ability to secrete insulin gradually; this phenotype is prevented by correction of the mutation. Our studies illuminate the molecular basis for the efficacy of treatment of HNF1A-MODY with sulfonylureas, and suggest promise for the use of cell therapies.

[1] Naomi Berrie Diabetes Center & Departments of Pediatrics and Medicine, Vagelos College of Physicians and Surgeons, Columbia University, New York, NY 10032, USA. [2] Institute of Human Nutrition, Columbia University Medical Center, New York, NY 10032, USA. [3] Department of Systems Biology, Columbia University Medical Center, New York, NY 10032, USA. [4] Department of Physiology and Cellular Biophysics, College of Physicians and Surgeons, Columbia University, New York, NY 10032, USA. [5] Stem Cell Core Facility, Department of Rehabilitation and Regenerative Medicine, Columbia University, New York, NY 10032, USA. [6] Center for Dementia Research, Nathan S. Kline Institute, Orangeburg, NY 10962, USA. [7] Department of Surgery, University of Virginia, Charlottesville, VA 22908, USA. [8] Columbia Center for Translational Immunology, Department of Surgery, Columbia University Medical Center, New York, NY 10032, USA. [9] Regeneron Pharmaceuticals, Tarrytown, NY 10591, USA. [10] Present address: Vertex Cell and Genetic Therapies, Watertown, MA 02472, USA. ✉email: de2220@cumc.columbia.edu

Maturity onset diabetes of the young (MODY) is an autosomal dominant form of diabetes with onset typically before the age of 25 years accounting for 1–5% of diabetes incidence[1,2]. There are at least 11 genes with mutations causing MODY, due to derangements in β-cell development or function. HNF1A-MODY is caused by mutations in the transcription factor HNF1A[3,4] and is among the most commonly diagnosed instances of MODY[5]. HNF1A-MODY patients have normal glucose tolerance during childhood and early adult life but show a progressive reduction of insulin secretion in response to glucose[6–9]. Glycemia typically increases over time, resulting in the need for treatment with insulin secretory sulfonylurea drugs. Eventually, 30–40% of patients become insulin-dependent due to progressive deterioration of β-cell function.

HNF1A is a 631-amino acid transcription factor with three major domains: dimerization, DNA-binding, and transactivation. Over 200 HNF1A diabetes-related mutations have been identified in all major ethnic groups[10]. Understanding the role of the HNF1A gene and the pathophysiology of HNF1A-MODY has been difficult because of the limited access to islets of affected individuals. Mouse models of HNF1A deficiency do not accurately mimic patient phenotypes[11].

Stem cell-derived β-cells provide a useful model system, and have been used to study β-cell development in humans[12,13] and to recapitulate disease phenotypes[14,15]. Differentiation of pluripotent stem cells to pancreatic endocrine cells can be achieved by a multistep protocol resulting in islet-like clusters containing all endocrine cell types[16–18]. Transplanting these islet-like clusters into mice enables maturation and functional testing of stem cell-derived β-cells (scβ-cells) in vivo[19,20]. Here we show that human stem cell-based models of HNF1A deficiency display islet developmental bias towards α-cells, altered insulin granule morphology and the stoichiometry of insulin:C-peptide secretion in vitro and in vivo. We used these models to identify disordered calcium homeostasis and accumulation of abnormal insulin granules as key mechanisms accounting for the secretory defects observed. This study was designed using two cell-based models: Human embryonic stem cell (hESC) lines rendered null for HNF1A in Figs. 1–5 and S1–S10; and induced pluripotent stem cell (iPSC) lines with HNF1A-MODY patient-specific mutations (R200Q) in Figs. 6 and S11–S13. Figure S1 provides a schematic overview of the study.

## Results

### Isogenic cell lines with HNF1A mutations in hESCs and HNF1A-MODY iPSCs.
To elucidate the cellular functions of HNF1A, we used CRISPR/Cas9 to generate hESC lines (Mel1) harboring non-naturally occurring heterozygous and homozygous null mutations. The Mel1 hESC line has a $INS^{GFP/wt}$ and $GAPDH^{Luciferase/wt}$ dual-reporter, enabling imaging and isolation of INS-GFP$^+$ cells[21]. GFP expression does not alter β-cell function in mice[22] and does not cause ER stress in human scβ-like-cells[23]. Short guide RNAs (sgRNAs) were used to target exon 3 of the HNF1A gene (Fig. S2a and Supplementary Data 1) because the DNA-binding domain in exon3 has the highest frequency of mutation in HNF1A-MODY patients[5]. Transfection of hESC lines with Cas9-GFP and sgRNAs #12 or #14, followed by sorting for GFP (Fig. S2b), achieved 17.3% and 20.4% indel efficiency as shown by surveyor assay (Fig. S2c). Gene editing resulted in compound heterozygous knockouts and heterozygous mutations (Fig. S2d) with no off-target mutations detected (Supplementary Data 2). Heterozygous (hESC HNF1A Het) and compound heterozygous-null mutant cell lines (hESC HNF1A KO1 and KO2) with premature protein termination were chosen for further studies (Fig. S2e). hESC HNF1A Het, KO1, and KO2 lines

retained the HNF1A dimerization domain (truncated HNF1A), while the hESC HNF1A KO3 line was mutated at the start codon, deleting the entire HNF1A protein (Fig. S2e).

To understand the consequences of HNF1A-MODY patient-specific heterozygous mutations, we also generated iPSC lines from HNF1A-MODY patient fibroblasts containing heterozygous mutations in the transactivation domain (HNF1A-MODY iPSC Het: +/460_461insCGGCATCCAGCACCTGC, ID1056); and DNA-binding domain (HNF1A-MODY iPSC Het: +/R200Q, ID1075 and ID1076). The R200Q variant has been previously associated with HNF1A-MODY[24]; the effect of this missense mutation is unknown, but likely pathogenic[24]. We corrected the R200Q mutation from HNF1A-MODY iPSC Het (Fig. S2f) with an efficiency of 7.9% (5 clones out of 63) to generate isogenic wild type cells. We also generated iPSCs compound heterozygous knockout lines with an efficiency of 42.9% (27 clones out of 63) (Fig. S2d). An iPSC line with a deletion of the WT allele, and the patient's mutation (HNF1A-MODY iPSC: R200Q/−) was chosen for further studies (Fig. S2g). We also introduced the same R200Q mutation into both alleles in the hESC line (hESC Hom: R200Q/R200Q) (Fig. S2h) with an efficiency of 20.8% (5 clones out of 24) (Fig. S2i). All genetically manipulated cell lines (Fig. S2j) resulted in modified versions of HNF1A protein (Fig. S2k, l), had a normal karyotype (Fig. S2m, n) and no mutations at potential off-target sites (Supplementary Data 2).

### HNF1A is not required to generate pancreatic endocrine cells.
To determine the functional consequences of HNF1A deficiency, hESC cells were differentiated into the pancreatic lineage[20] (Fig. S3a). Differentiation of wild type stem cells consistently generated 82% PDX1$^+$/NKX6.1$^+$ cells at the pancreatic progenitor stage (day 11). At the endocrine stage (day 27), we obtained 45% PDX1$^+$/CPEP$^+$ cells with 60% of them co-expressing NKX6.1 (scβ-like cells), 30% GCG$^+$ cells (scα-like cells), and 10% SST$^+$ cells (scδ-like cells) (Fig. S3b).

To determine the timing of HNF1A expression, we performed qPCR at different stages of differentiation. Insulin mRNA was detected at the endocrine stage (day 20) (Fig. S3c), whereas HNF1A mRNA is first detected at the primitive gut tube stage (day 5), increased at the pancreatic progenitor stage (day 11) and increased further at the endocrine stage (day 27) (Fig. S3d). HNF1A knockout resulted in a significant reduction of HNF1A transcript at the endocrine stage (Fig. S3e); HNF1A protein was detected in hESC WT, but only a faint signal was detected in hESC HNF1A KO-derived endocrine cells by western blot (Fig. S3f).

Mutations in HNF1A did not affect hESC lines in generating definitive endoderm cells (SOX17$^+$) at day 3 or pancreatic progenitor cells (PDX1$^+$/NKX6.1$^+$) at day 11 of differentiation (Fig. S3g). At the endocrine stage (day 27), organoids were morphologically indistinguishable by the HNF1A genotype and showed no differences in INS-GFP fluorescence (Fig. S3h). HNF1A KO cells were capable of differentiation to all islet endocrine cell types, including PDX1$^+$/NKX6.1$^+$/SYP$^+$/CPEP$^+$ cells (scβ-like cells), GCG$^+$ cells (scα-like cells), and SST$^+$ cells (scδ-like cells) (Figs. 1a and S3i–m), indicating that HNF1A is not required to generate pancreatic endocrine cells. No differences were found for PDX1 and NKX6.1 in hESC HNF1A WT and HNF1A KO endocrine cells (day 27) by immunohistochemistry (Fig. S3j, k).

### HNF1A deficiency impairs a network of genes required for calcium signaling, glucose-stimulated insulin secretion, and β-cell fate.
To determine the transcriptional consequences of HNF1A deficiency in scβ-cells, bulk and single-cell RNA sequencing was performed in INS-GFP sorted β-like cells derived from hESC

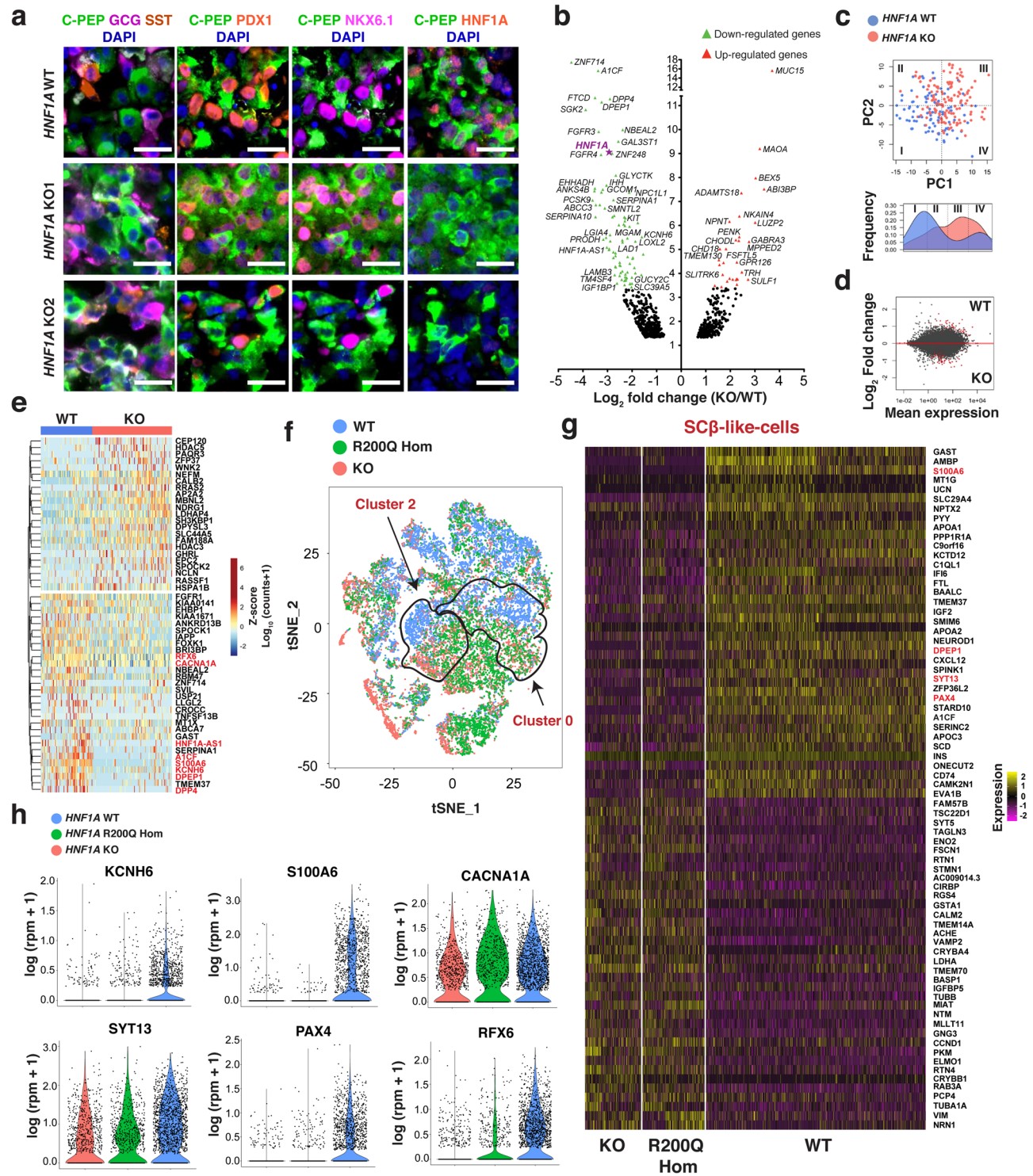

HNF1A KO and WT lines in vitro. Volcano plot analysis of bulk RNA sequencing identified 30 up-regulated genes and 148 down-regulated genes in *HNF1A* KO β-like cells (Fig. 1b and Supplementary Data 3). Single-cell RNA sequencing showed that *HNF1A* KO β-like cells segregated from *HNF1A* WT β-like cells based on their gene expression profile by principal component analysis (Fig. 1c), and volcano plot analysis (Fig. 1d) revealed previously undescribed HNF1A target genes involved in intracellular calcium signaling (*CACNA1A, S100A6,* and *TMEM37*), calcium-mediated insulin secretion (*KCNH6*) and glucose-stimulated insulin secretion (*RFX6, DPEP1, DPP4,* and *IAPP*) (Fig. 1e) as commonly

downregulated with bulk RNA sequencing from *HNF1A* KO scβ-like cells. Among the up-regulated genes in *HNF1A* KO β-like cells, we found genes involved in the synaptic vesicle cycle (*AP2A2, SH3KBP1,* and *HSPA1B*) (Fig. 1e and Supplementary Data 4).

To understand the transcriptional consequences of *HNF1A* mutation in other endocrine cell types, we analyzed all single cells within the islet-like clusters. In addition to *HNF1A* KO lines, an hESC line homozygous for *HNF1A* R200Q (R200Q/R200Q) was included. Using clustering analysis from those cells, we grouped cells into 13 different populations (Fig. S4a, b). To cluster different endocrine cells, we used the expression of endocrine cell

**Fig. 1 HNF1A deficiency impairs a network of genes required for calcium signaling, glucose-stimulated insulin secretion, and β-cell fate in vitro.**
**a** Representative immunohistochemistry (IHC) images of hESC-derived endocrine cell lines (*HNF1A* WT, KO1, and KO2) for indicated markers. White cells are GCG/CPEP double-positive cells. Scale bars: 20 µm. **b–e** RNA sequencing of FACs sorted *INS^GFP/wt* positive cells between *HNF1A* WT and KO genotypes (*n* = 3 for both genotypes). **b** Volcano plot depicting fold change (log₂ fold change, *x*-axis) and statistical significance (−log₁₀ *p*-value, *y*-axis) for differential gene expression (down-regulated in green; up-regulated in red) by bulk RNAseq (see also Supplementary Data 3). **c** Cell clustering by principal component analysis (PCA). The lower panel depicts the radial distribution or frequency of individual cells from each PCA quadrant. **d** MA-plot depicting fold change (log₂-fold change, *y*-axis) and mean expression (counts, *x*-axis) of differentially expressed genes. **e** Heatmap showing expression for each gene identified as *z*-score of expression from all sorted scβ-like cells. **f** Single-cell RNA sequencing of 22,164 (all genotypes combined) unsorted hESC-derived endocrine cells by *HNF1A* WT, R200Q homozygous, and KO genotypes (*n* = 3 for each genotype). Feature plot based on tSNE projection of cells where the colors denote different cell lines by *HNF1A* genotype line via Louvain algorithm performed by Seurat. **g** Heatmap showing differentially expressed genes from scβ-like cells by *HNF1A* WT, R200Q homozygous, and KO genotypes. Total of 1846 scβ-like cells (all genotypes combined) were identified and displayed as |log FC | >0.35 and adjusted *p*-value < 1e⁻⁴. Genes are listed in decreasing order of log₂-fold change between *HNF1A* WT and *HNF1A* mutant genotypes. **h** Violin plot of cells based on *KCNH6, S100A6, CACNA1A, SYT13, PAX4,* and *RFX6* gene expression (log1(rpm + 1)) from scβ-like cells. All stem cell differentiation was done for 27 days. (*n*) represents the number of biological replicates. See also Figs. S4 and S5.

markers *SYP*, *INS*, *GCG,* and *SST* (Fig. S4c) and pancreatic progenitor markers *NKX6.1*, *PDX1*, *MAFA*, and *HNF1A* (Fig. S4d). This allowed the identification of insulin-expressing cells (cluster 0) and glucagon-expressing cells (cluster 2) (Fig. 1f). From the population of insulin-expressing cells (cluster 0), cells defined by high insulin and low glucagon expression were considered to be scβ-like cells and expression was compared between genotypes (Fig. S5a).

In scβ-like cells, a total of 73 genes were differentially expressed between *HNF1A* WT, homozygous R200Q, and KO lines (Fig. 1g and Supplementary Data 5). Pathway analysis of differentially expressed genes in scβ-like cells revealed genes involved in MODY, endocrine cell development, calcium signaling/sensing, insulin secretion and synaptic vesicle cycle (Fig. S5b). Furthermore, several MODY genes (*RFX6*, *PAX4* and *NEUROD1*) were down-regulated (Fig. 1g). These transcription factors are important for the identity of adult pancreatic β-cells[25,26]. We also identified down-regulated genes important for intracellular calcium signaling (*S100A6* and *TMEM37*), exocytosis-regulated insulin secretion (*SYT13*), and glucose-stimulated insulin secretion (*IGF2* and *LLGL2*) in *HNF1A*-mutated β-like cells (Fig. 1g and Supplementary Data 5)[25,27,28]. Among the up-regulated genes, we found genes involved in the synaptic vesicle cycle (*RAB3A* and *VAMP2*) (Fig. 1g and Supplementary Data 5). *RFX6*, *KCNH6*, *S100A6*, *A1CF*, *DPEP1*, and *DPP4* were the most consistently downregulated genes in *HNF1A* mutated β-like cells. In addition to those genes, *SYT13*, *CACNA1A*, and *PAX4* were down-regulated in the scβ-like-cell subpopulation (Fig. 1h). A recent case study of cadaveric human islets of an HNF1A-MODY donor (+/T260M) also found down-regulation of *RFX6*, *CACNA1H*, *IAPP*, and *TMEM37* in β-cells, and down-regulation of *CACNA1A*, *RFX6*, *PCBD1*, and *PPP1R1A* in α-cells[29]. These genes are required for calcium signaling, glucose-stimulated insulin secretion, and determination of endocrine cell fate.

To determine the molecular consequences of *HNF1A* deficiency in other endocrine cell types, we analyzed mono-hormonal scα-like cells (cluster 2a), characterized by high glucagon and low insulin expression (Fig. S5c, d). We identified up-regulated genes involved in glucagon signaling pathways (*PKM*, *CALM1*, and *CALM2*) and down-regulated genes involved in insulin secretion (*ADCY1*, *KCNH6*, *RFX6*, *IGF2*, *TM4SF4* and *ATF4*), calcium-mediated exocytosis (*SYT7*) and β-cell dedifferentiation (*GC*)[30] (Fig. S5e, f). Furthermore, in bi-hormonal cells (cluster 2b) expressing both insulin and glucagon, *PYY* was down-regulated, while *GCG* was up-regulated (Fig. S5g, h) in *HNF1A* mutated lines.

A recent publication by Cardenas-Diaz et al.[31] identifies *LINC01139* as an *HNF1A* target implicated in β-cell respiration and mitochondrial function. We found no significant difference in the expression of this long non-coding RNA between WT and *HNF1A*

mutated (hESC Het, hESC Hom R200Q/R200Q, hESC KO or iPSC HNF1A-MODY Het +/460ins, ID1056) β-like (Fig. S5i–k) and α-like cells (hESC Hom R200Q/R200Q and hESC KO) (Fig. S5l). No difference was found in cadaveric human β-cells from non-diabetic donors as compared to one HNF1A-MODY donor (+/T260M) (Fig. S5m).

In summary, *HNF1A* orchestrates a network of genes promoting endocrine differentiation to β-cells, and in differentiated β-cells regulates genes involved in calcium signaling, hormone exocytosis, and glucose-stimulated insulin secretion.

**HNF1A deficiency causes a developmental bias towards the α-cell fate.** Downregulation of genes involved in adult β-cell identity (*PAX4* and *RFX6*) with upregulation of glucagon in *HNF1A* KO cells point to a developmental role of HNF1A. Both hESC *HNF1A* KO and WT cells gave rise to insulin-positive cells (Fig. S6a, b), with increased glucagon-positive cells in the hESC *HNF1A* KO line (Figs. 2a and S6b). This finding was confirmed by an increase in GCG⁺/CPEP⁺ cell type ratio by immunohistochemistry (Fig. 2b) and glucagon content by ELISA (Fig. 2c). The increase in scα-like cells in *HNF1A* KO cells was driven by an ~2-fold increase in the number of bi-hormonal scβ/α-like cells as shown by flow cytometry (Figs. 2d and S3m), immunohistochemistry (Figs. 2b and S6c), and RNA sequencing (Fig. S6d). Transcriptional analysis by single-cell RNA sequencing of glucagon and insulin double hormone-positive cells showed an increase in glucagon gene expression and a decrease in insulin gene expression in *HNF1A* KO cells compared to WT cells (Fig. 2e). These findings are consistent with *PAX4* repressing pancreatic glucagon gene expression[32].

To assess the developmental requirements of *HNF1A* in endocrine cells in vivo, we transplanted pancreatic clusters derived from hESC *HNF1A* WT and hESC *HNF1A* KO lines with *GAPDH^Luciferase/wt* and *INS^GFP/wt* dual-reporter[21]. Mice received ~180 clusters of stem cell-derived islet-like cells (Fig. S7a) containing a similar amount of CPEP/PDX1 positive cells across genotypes (Fig. S7b, c). Cell transplantation was done by injection into the ventral and medial muscle groups of the left quadriceps in NOD SCID gamma immunodeficient mice (NSG mice) (Fig. 2f). Skeletal muscle has been used for other endocrine transplants, including for parathyroid auto-transplantation in patients undergoing parathyroidectomy with a 93% success rate[33]. Analysis of engraftment efficiency was evaluated by bioluminescence intensity (BLI). Mice with successful engraftment showed a two-fold increase in BLI 4–6 weeks post-transplantation, while those with failed engraftment showed a decrease (Fig. S7d). Transplantation was successful in 79% (31/39) of mice, independent of the *HNF1A* genotype (Fig. S7e). 92.5% (49/53) of mice transplanted remained teratoma-free

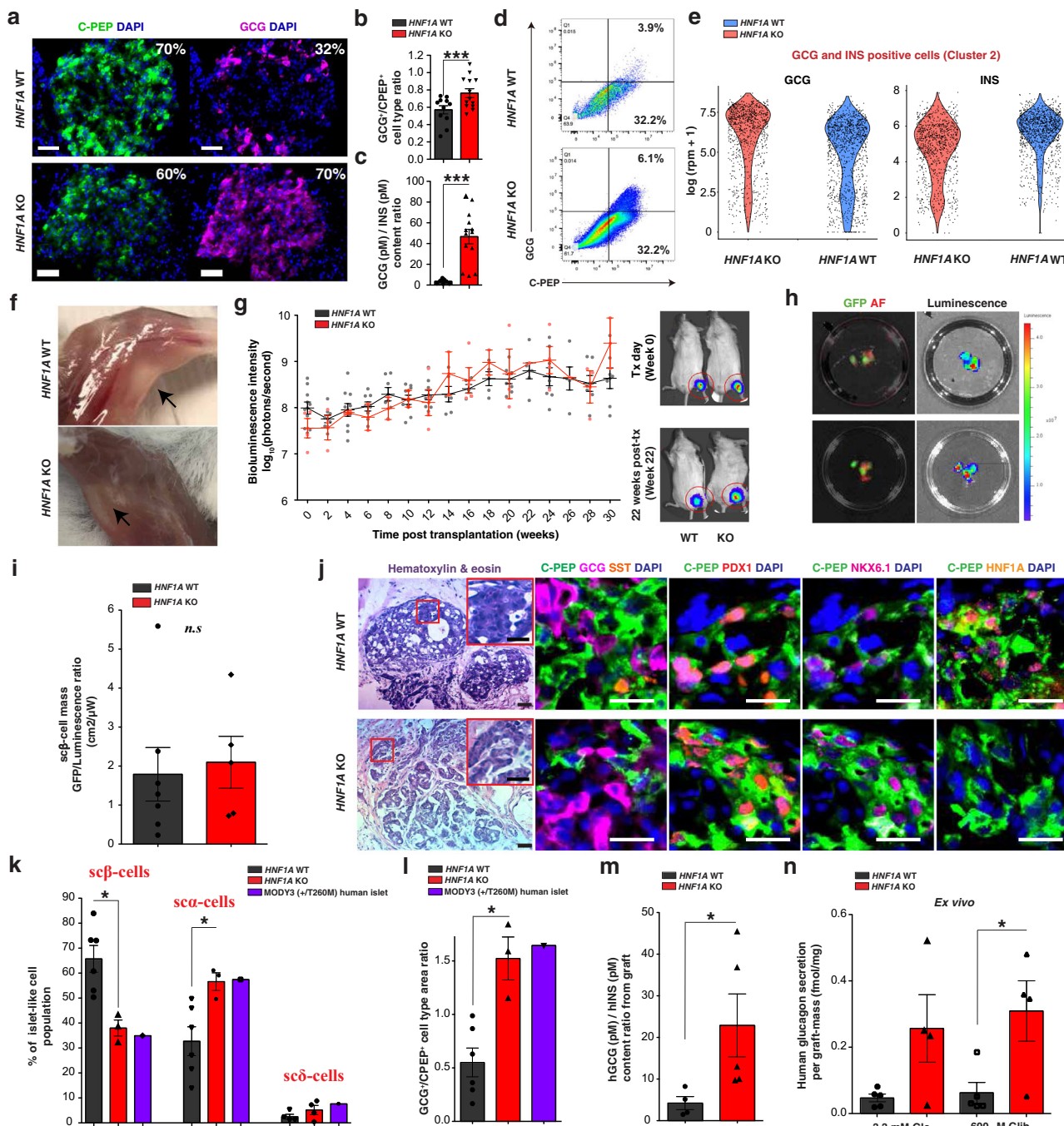

(Fig. S7f). In those mice, the graft explant (30 weeks post-transplantation) (Fig. S7g) was a vascularized tissue of about ~220 mg (Fig. S7h). According to luminescence intensity, there was a gradual increase in BLI from week 0 to 30 post-transplantation without significant differences by *HNF1A* genotype (Fig. 2g). Grafts remained localized for up to 50 weeks post-transplantation, and in no case (0/39) was luminescence detected in an ectopic location. These results show that skeletal muscle is a stable and suitable transplantation site for SC-derived islet-like cells.

Thirty weeks post-transplantation, we isolated grafts from normoglycemic animals. Quantification of scβ-mass as determined by GFP fluorescence did not differ by *HNF1A* genotype (Figs. 2h, i and S7i, j). The presence of exclusively mono-hormonal endocrine cells, including scβ-cells (CPEP[+]/PDX1[+]/

NKX6.1[+]), as well as glucagon- and somatostatin-positive cells were apparent in both hESC *HNF1A* WT and hESC *HNF1A* KO grafts (Fig. 2j). No double hormone-positive cells were identified in either genotype. Consistent with the in vitro studies, the percentage of scα-cells in *HNF1A* KO compared to *HNF1A* WT islet-like structures was increased by 24% (Figs. 2k and S7k), leading to a significant increase in GCG[+]/CPEP[+] cell type ratio (Fig. 2l); comparable to cadaveric human islets of an HNF1A-MODY donor (+/T260M)[29] (Fig. 2k, l). Ex vivo analysis of isolated *HNF1A* KO grafts showed an increase in glucagon protein content by ELISA (Figs. 2m and S7l, m). Glucagon secretion was higher in *HNF1A* KO grafts when stimulated with glibenclamide, a second-generation sulfonylurea drug compared to *HNF1A* WT grafts (Fig. 2n). Similarly, elevated glucagon secretion upon KCl stimulation was detected in vitro from

**Fig. 2 HNF1A deficiency causes a developmental bias toward the α-cell fate. a** Representative IHC images of hESC-derived endocrine cell lines (*HNF1A* WT and KO) for indicated markers. Scale bars: 50 µm. **b** GCG⁺/CPEP⁺ cell type ratios by IHC and **c** GCG/INS protein content ratios measured by ELISA. *p*-values: ****p* < 0.001; two-tailed *t*-test. **d** CPEP⁺ and GCG⁺ populations by flow cytometry for hESC-derived endocrine cells (*HNF1A* WT and KO). Gating for GCG and CPEP-negative cells was determined by incubating hESC cells (negative control) with primary and with secondary antibodies. **e** Violin plot of total of 2670 (all genotypes) GCG and INS positive cells (cluster 2) displayed based on *GCG* and *INS* expression (log(rpm + 1)) (*n* = 3 for each genotype). **f** Representative image of graft tissue (black arrows) 30 weeks post-transplantation with hESC-derived endocrine cells before explant from quadriceps muscle. **g** Bioluminescence intensity (log₁₀ photons/s) measured in mice transplanted with hESC-derived endocrine cells (WT *n* = 11 and KO *n* = 4) over time (weeks) with representative bioluminescence images. **h** Representative GFP fluorescence (scβ-cells) and bioluminescence images (total cells) (GFP in green; AF: tissue auto-fluorescence in red) from isolated grafts with **i** quantification of scβ-cell mass GFP/luminescence ratio (cm²/µW). **j** Representative H&E and IHC images showing hESC-derived endocrine cells from isolated grafts for indicated markers. Scale bars: 20 µm. **k** Percentage of scβ-, scα- and scδ-cells from the islet-like clusters and HNF1A-MODY islets (*HNF1A* +/T260M) from Haliyur et al. ²⁹. **l** GCG⁺/CPEP⁺ cell type area ratios by IHC from isolated grafts and GCG⁺/INS⁺ from HNF1A-MODY islets (*HNF1A* +/T260M)²⁹. Each point in plot is an islet-like cluster. Endocrine cell percentages were calculated as the hormone-positive areas from islet-like clusters across the entire graft (2 grafts for each genotype). **m** hGCG (pM)/hINS (pM) content ratio from isolated grafts. **n** Ex vivo human glucagon secretion normalized to graft mass (fmol/mg) in response to indicated secretagogues. All protein concentrations were measured by ELISA. All stem cell differentiation was done for 27–30 days. 20 clusters (~10k cells per cluster) of endocrine cells were used flow cytometry. All mice were transplanted with clusters of hESC-derived endocrine cells (*HNF1A* WT and KO2) at day 27 of differentiation, and grafts were isolated 30 weeks post-transplantation for ex vivo analysis. *n* Represents the number of biological replicates. For scatter plots, each point in the plot represents an independent biological experiment (*n*). Data are represented as mean ± SEM. *p*-values: **p* < 0.05, ***p* < 0.01, ****p* < 0.001; Mann–Whitney test. n.s. non-significant. See also Figs. S6 and S7.

*HNF1A* KO-derived endocrine cells (Fig. S7n). These results point to both a gain of α-cell number and enhanced α-cell function due to *HNF1A* deficiency. This is consistent with the up-regulation of genes (*PKM*, *CALM1*, and *CALM2*) involved in glucagon signaling pathways in scα-like cells.

**HNF1A deficiency affects glucose-mediated secretion of insulin.** To determine the consequences of *HNF1A* deficiency on β-cell function, we measured glucose-stimulated insulin secretion. *HNF1A* KO scβ-like cells had reduced basal insulin secretion (Fig. S8a) and impaired glucose-stimulated insulin secretion compared to WT scβ-like cells and compared to human pancreatic islets in vitro (Fig. 3a). These differences in hormone secretion between *HNF1A* mutant and *HNF1A* WT scβ-like cells were not due to a reduction in insulin content (Fig. S8b). In contrast to glucose, treatment with the insulin secretagogues, tolbutamide, or potassium chloride resulted in insulin secretion in *HNF1A* KO cells comparable to *HNF1A* WT β-like cells (Fig. 3b). Thus, membrane depolarization by sulfonylurea causes secretion of insulin from *HNF1A* mutant β-cells that do not respond to glucose alone, consistent with the clinical efficacy of sulfonylurea drugs. These differences in response to insulin secretagogues are likely intrinsic to the β-cell and not due to augmented glucagon³⁴.

As calcium signaling is a mechanism for the release of insulin, we measured intracellular calcium levels in dispersed scβ-like cells in vitro. Both genotypes had glucose-stimulated calcium responses (Fig. 3c). However, hESC *HNF1A* KO β-like cells had significantly lower calcium levels relative to hESC *HNF1A* WT scβ-like cells as measured by Fura-2 fluorescence (Fig. 3c). This finding was further supported by measurements of absolute intracellular calcium concentrations by Fura-2 fluorescence (Fig. 3d). Among the HNF1A target genes, reduced expression of *CACNA1A* is potentially involved in reducing intracellular calcium levels, and *SYT13* may be required for efficient exocytosis-mediated insulin secretion. *CACNA1A* encodes a voltage-dependent calcium channel mediating the entry of calcium and is involved in calcium-dependent insulin secretion and type 2 diabetes³⁵. Synaptotagmins are calcium sensors localized in the β-cell insulin granules and are required for vesicle fusion and glucose-stimulated insulin release³⁶⁻³⁸. *SYT13* is a member of the synaptotagmin family and is predicted to be involved in calcium-regulated exocytosis. *SYT13* is down-regulated in human T2D islets and silencing of *SYT13* impairs insulin secretion in INS1-832/13 cells³⁹. To determine whether the down-regulation of *CACNA1A*

and *SYT13* seen in *HNF1A* mutant cells (Fig. 1h) reproduced the reduced intracellular calcium levels and reduced glucose-stimulated insulin secretion, we generated hESC *CACNA1A* KO and *SYT13* KO cell lines using CRISPR/Cas9. Sanger sequencing revealed homozygous mutations resulting in frame shifts (Fig. S8c). Differentiation of *CACNA1A* KO and *SYT13* KO lines generated INS-GFP organoids (Fig. 3e) comprised of ~51% PDX1/CPEP and ~5% GCG positive cells (Figs. 3f and S8d) with no differences compared to isogenic WT cells. Dispersed *CACNA1A* KO β-like cells (Fig. S8e) had significantly reduced intracellular calcium levels compared to WT cells (Figs. 3g and S8f). The reduction of intracellular calcium in *CACNA1A* KO was intermediate to *HNF1A* KO cells, indicating that *CACNA1A* is not the only gene affecting intracellular calcium levels in *HNF1A* KO cells. Intracellular calcium levels in *SYT13* KO were unchanged compared to WT cells (Figs. 3g and S8f). Reduced basal and stimulated insulin secretion were observed in both *CACNA1A* KO and *SYT13* KO scβ-like cells at intermediate levels to *HNF1A* KO scβ-like cells (Fig. 3h). Treatment with tolbutamide or KCl stimulated insulin secretion in all KO lines comparable to the corresponding WT lines. Impairments in insulin release due to reduced levels of these molecules can be overcome by elevating calcium beyond physiological levels using tolbutamide or KCl. The milder phenotypes of cells deficient for single target genes compared to *HNF1A* KO cells suggests a combined effect on several HNF1A target genes on insulin secretion.

**HNF1A deficiency alters the stoichiometry of insulin to C-peptide secretion.** To interrogate the function of *HNF1A* KO β-cells in vivo, we monitored circulating human insulin and C-peptide concentrations in euglycemic mice transplanted with sc-islet-like clusters. By 4 weeks post-transplantation, plasma human C-peptide concentrations were significantly lower in mice transplanted with *HNF1A* KO β-cells compared to *HNF1A* WT β-cells (Fig. 4a). Circulating human C-peptide concentrations in mice transplanted with *HNF1A* WT β-cells increased gradually, reaching human physiological levels (652 ± 146 pM) 24–30 weeks post-transplantation. In mice transplanted with *HNF1A* KO β-cells, plasma human C-peptide concentrations increased to a maximum of 352 ± 142 pM at 30 weeks post-transplantation (Fig. 4a). In mice transplanted with *HNF1A* WT β-cells, human insulin was detected as early as 4 weeks post-transplantation and concentrations increased over time, reaching human physiological levels (12.74 ± 2.1 mU/L) 24–30 weeks post-transplantation.

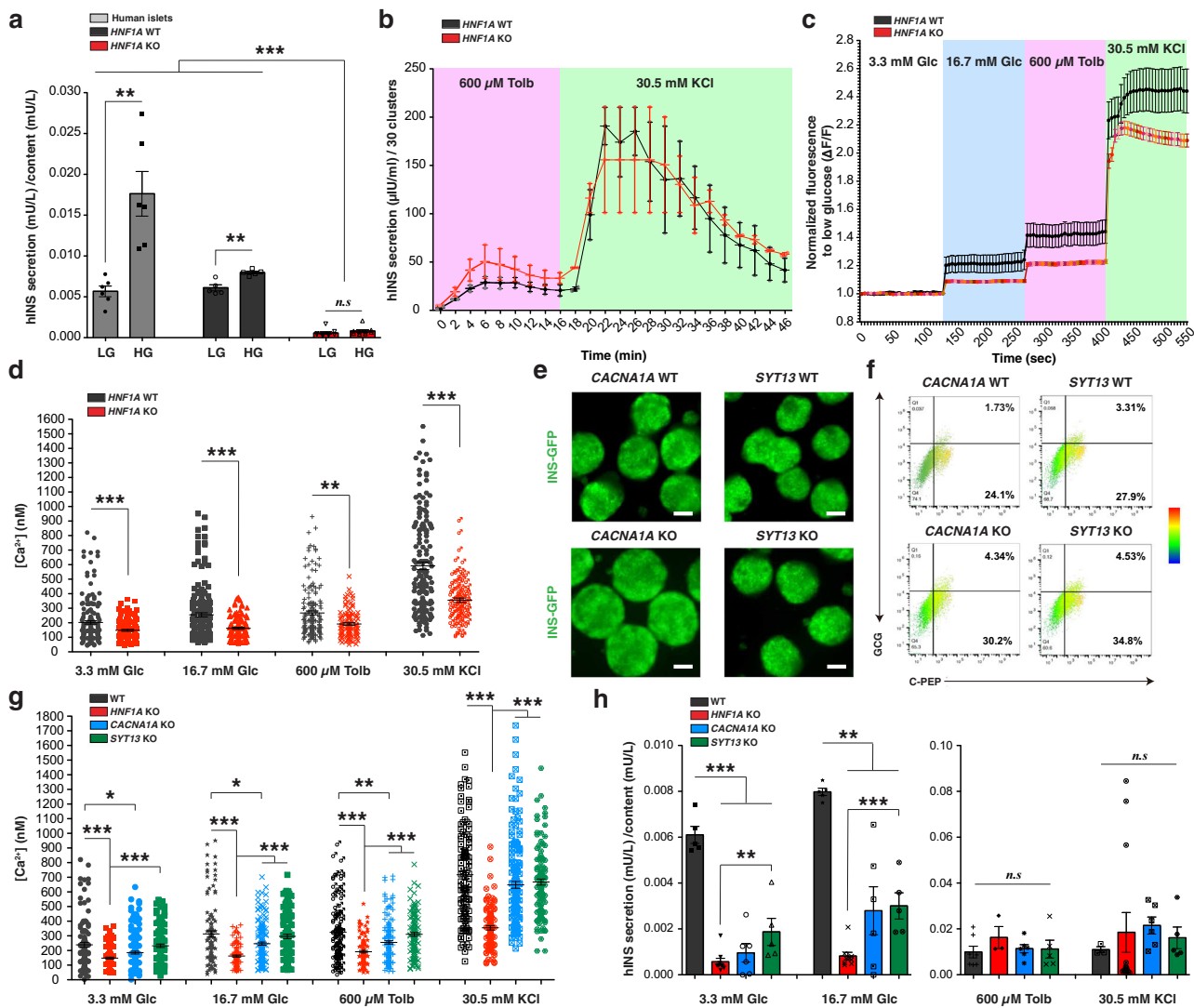

**Fig. 3 HNF1A deficiency affects insulin secretion due to reduced intracellular calcium levels in association with CACNA1A and SYT13 down-regulation in vitro. a** Human insulin secretion (mU/L) in 1 h normalized to content (mU/L) in response to low glucose (LG, 3.3 mM) and high glucose (HG, 16.7 mM) stimulation in the static assay of hESC-derived endocrine cell lines (*HNF1A* WT and KO) and pancreatic human islets. **b** Human insulin secretion (μIU/ml) normalized to 30 clusters in a perfusion assay (both *HNF1A* WT and KO genotypes n = 2) in response to indicated secretagogues. **c** Intracellular calcium fluorescence normalized to low glucose (F340/380) from dispersed scβ-like cells with **d** absolute intracellular calcium concentration (nM) quantification in response to indicated secretagogues (each point represents an scβ-like-cell, *HNF1A* WT n = 170 and *HNF1A* KO n = 110 cells). **e** Representative GFP-field images of endocrine organoids from indicated genotypes. Scale bars: 200 μm. **f** Representative CPEP⁺ and GCG⁺ populations quantification by flow cytometry. **g** Absolute intracellular calcium concentration (nM) in response to indicated secretagogues (each point represents a scβ-like-cell from independent batches of differentiation, WT n = 110, *HNF1A* KO n = 110, *CACNA1A* KO n = 175, and *SYT13* KO n = 120 cells). **h** Human insulin secretion (mU/L) in 1 h normalized to content (mU/L) in response to low (3.3 mM) and high glucose (16.7 mM) and indicated secretagogues in static assay (WT n = 5, *HNF1A* KO n = 9, *CACNA1A* KO n = 6 and *SYT13* KO n = 5). All stem cell differentiation was done for 27 days. 20 clusters (~10k cells per cluster) of endocrine cells were used flow cytometry. *n* represents the number of biological replicates. For scatter plots, each point in the plot represents an independent biological experiment (*n*). Data are represented as mean ± SEM. *p*-values: *p < 0.05, **p < 0.01, ***p < 0.001; Mann–Whitney test. n.s. non-significant. See also Fig. S8.

However, mice transplanted with hESC *HNF1A* KO β-cells had virtually undetectable plasma human insulin concentrations for 30 weeks (Fig. 4b) despite human C-peptide levels of >300 pM. These differences in hormone secretion between *HNF1A* mutant and WT scβ-cells were not due to reduced scβ-cell mass in vivo (Fig. 2i). These findings led us to quantify the secretion ratio of hormones in scβ-cells in vitro. For equivalent secretion of human C-peptide at basal glucose condition (Fig. S8g), secretion of human insulin was significantly reduced in hESC *HNF1A* KO lines (Fig. S8h), leading to decreased insulin:C-peptide secretion ratios in vitro (Fig. S8i). Thus, the absence of insulin in the

plasma was not a limitation of the sensitivity of the assay but a decrease in the stoichiometry of insulin to C-peptide in circulation. While plasma of mice grafted with *HNF1A* WT β-cells consistently showed an insulin:C-peptide molar ratio of 0.22 ± 0.11 from week 4 to 30 post-transplantation, mice grafted with *HNF1A* KO β-cells had an 18-fold lower ratio (Fig. 4c). This decrease in circulating insulin:C-peptide ratio was reciprocal to a 3-fold increase in the insulin:C-peptide ratio of intracellular content from *HNF1A* KO isolated grafts (Fig. 4d), indicating a complementary imbalance of insulin to C-peptide. The altered insulin:C-peptide ratio is not attributable to differences in insulin

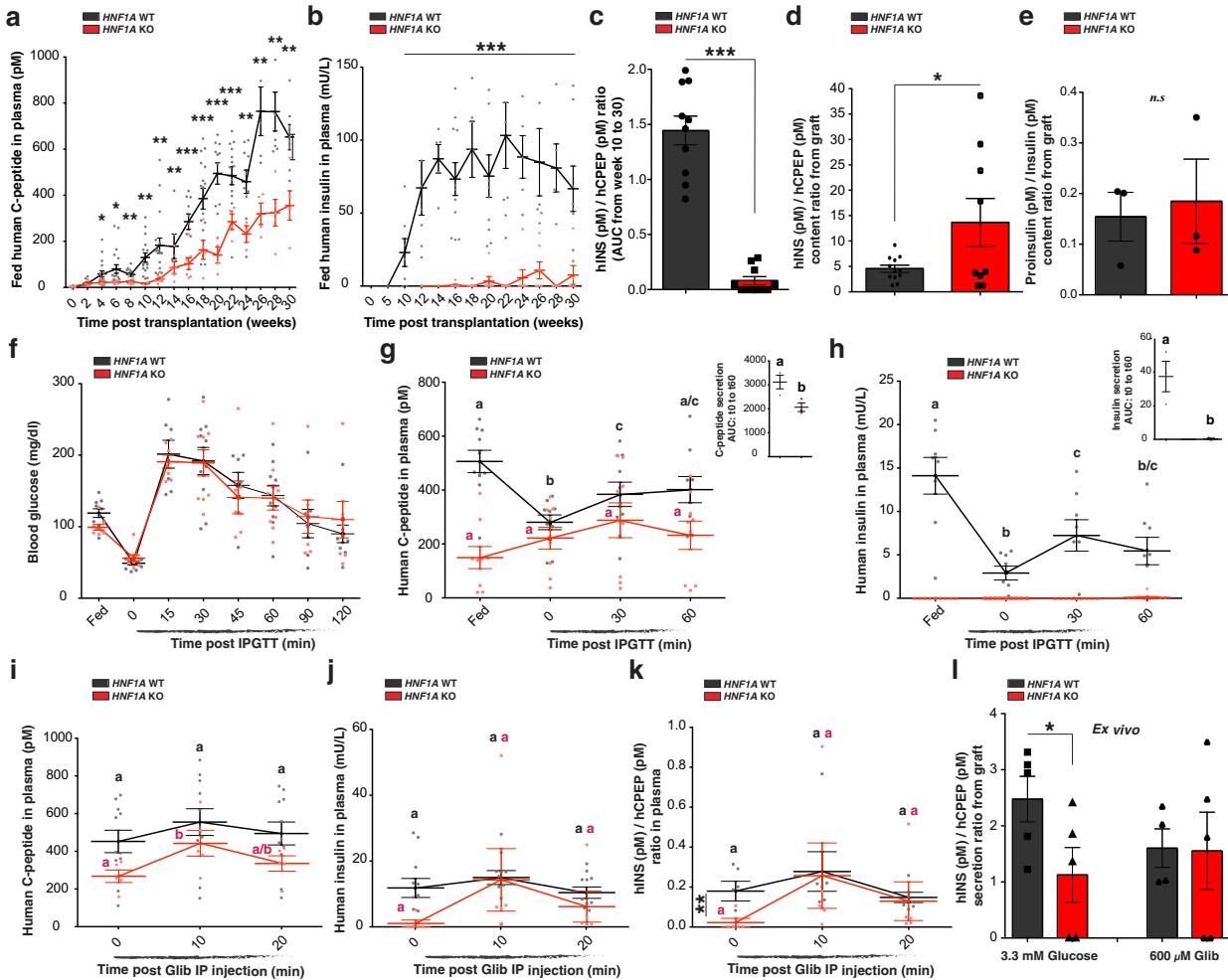

**Fig. 4 HNF1A deficiency alters the stoichiometry of insulin to C-peptide secretion in vivo. a** Plasma human C-peptide (pM) and **b** human insulin secretion (mU/L) in plasma of ad libitum-fed mice transplanted with hESC-derived endocrine cells (*HNF1A* WT $n = 18$ and KO $n = 7$). **c** hINS(pM)/hCPEP(pM) ratios are displayed as the area under the curve (AUC) from week 10 to week 30 post-transplantation. **d, e** Content ratio from isolated grafts: **d** hINS(pM)/hCPEP(pM) content ratio and **e** Proinsulin(pM)/Insulin(pM) content ratio. **f–h** IPGTT in mice transplanted with hESC-derived endocrine cells (*HNF1A* WT $n = 10$ and KO $n = 8$) in ad libitum-fed state and during an iPGTT (t0, t30 and t60). **f** Blood glucose concentrations (mg/dl), **g** human C-peptide (pM), and **h** human insulin secretion (mU/L) in plasma. AUC: area under the curve from t0 to t60. *p*-values were **b**: $p < 0.001$, **c**: $p < 0.05$, **a**, **c**: $p < 0.05$ and **b**, **c**: $p < 0.01$; two-tailed *t*-test. **i–k** Glibenclamide injection in ad libitum-fed mice transplanted with hESC-derived endocrine cells (*HNF1A* WT $n = 6$ and KO $n = 5$). **i** Human C-peptide secretion (pM), **j** human insulin secretion (mU/L), and **k** hINS (pM)/hCPEP (pM) secretion ratios. *p*-values were **b**: $p < 0.05$. **l** hINS(pM)/hCPEP(pM) secretion ratios ex vivo in response to indicated secretagogues. All mice were transplanted with clusters of hESC-derived endocrine cells at day 27 of differentiation, and grafts were isolated 30 weeks post-transplantation for ex vivo analysis. All protein concentrations were measured by ELISA. *n* represents the number of biological replicates. For scatter plots, each point in the plot represents an independent biological experiment (*n*). Data are represented as mean ± SEM. Different letters designate significant differences within the group. *p*-values: \**p* < 0.05, \*\**p* < 0.01, \*\*\**p* < 0.001; two-tailed *t*-test. n.s. non-significant. See also Fig. S8.

processing because insulin:proinsulin ratios in *HNF1A* KO and *HNF1A* WT grafts were identical in vivo (Fig. 4e) and in vitro (Fig. S8j–l), and no differences were found in the transcript levels of processing genes (*PC1/PC3*) (Supplementary Data 3–5). Impaired stoichiometry of circulating insulin to C-peptide was also observed in mice transplanted with two additional homozygous-mutant hESC lines (Fig. S8m). Therefore, *HNF1A* deficiency not only impairs insulin secretion, but also the stoichiometry of circulating insulin to C-peptide ratios.

To evaluate glucose-stimulated insulin secretion, we performed intraperitoneal glucose tolerance tests (iPGTT). During fasting and following an iPGTT, mice transplanted with *HNF1A* WT β-cells displayed normal human insulin and C-peptide secretion profiles (Fig. 4f–h). In contrast, *HNF1A* KO β-cells were non-responsive to glucose: plasma human C-peptide was not decreased by fasting and

showed no significant increase after glucose injection (Fig. 4g). Circulating human insulin remained undetectable in animals engrafted with *HNF1A* KO β-cells (Fig. 4h). Therefore, *HNF1A* deficiency affects glucose-stimulated insulin secretion, and C-peptide is released independent of metabolic state, resulting in an altered ratio of insulin to C-peptide.

**Glibenclamide restores insulin secretion in *HNF1A*-deficient scβ-cells.** In contrast to glucose, intra-peritoneal injection of the sulfonylurea drug, glibenclamide, in mice transplanted with hESC *HNF1A* KO β-cells resulted in a significant increase in plasma C-peptide after 10 min (Fig. 4i). In parallel, plasma human insulin concentrations in mice transplanted with *HNF1A* KO β-cells increased from undetectable levels to $14 \pm 21$ mU/L, reaching

levels comparable to the control group (Fig. 4j). Insulin:C-peptide ratios in plasma of animals transplanted with *HNF1A* KO β-cells were increased 10-fold by glibenclamide, equal to the ratios in mice transplanted with *HNF1A* WT β-cells (Fig. 4k). Clearance of insulin from the circulation occurred with the same kinetics in mice grafted with *HNF1A* WT and *HNF1A* KO β-cells 20 min after glibenclamide administration (Fig. 4k), excluding insulin stability or increased clearance in plasma as the cause for the low insulin concentrations in mice transplanted with *HNF1A* mutant scβ-cells. To further test whether reduced insulin:C-peptide secretion ratios from *HNF1A* KO β-cells were due to secretory defects and not insulin stability or clearance, we isolated the *HNF1A* KO grafts from mice and identified a significant decrease in insulin:C-peptide secretion ratios compared to *HNF1A* WT grafts ex vivo. This ratio was restored after exposure of the grafts to glibenclamide (Fig. 4l). Thus, membrane depolarization initiated by the closure of the ATP-sensitive K$^+$ (K$_{ATP}$) channels (target of glibenclamide) and high intracellular calcium levels cause the secretion of insulin that is abnormally retained in *HNF1A*-mutant β-cells.

### *HNF1A* deficiency causes abnormal insulin granule structure.
To identify the basis of the insulin secretory defect in *HNF1A* KO β-cells we examined insulin granule morphology by electron microscopy from 30-week explants of normoglycemic mice. *HNF1A* WT grafts contained scβ-cells with granules of a high-electron density core separated from limiting membranes by a "halo" (Figs. 5a and S9a), similar to human β-cells. In contrast, *HNF1A* KO islets-like scβ-cells showed a majority (>90%) of insulin granules that were abnormally enlarged (Fig. 5a–c) with a diffuse electron-light core (Figs. 5d, e and S9a), characteristic of immature granules. Enlarged insulin granules were further confirmed by immunogold staining for human C-peptide (Fig. 5f). Similar results were observed in vitro, where *HNF1A*-mutated cells had increased insulin granules diameters with a diffuse light core (Fig. S9b, c). For scα-cells, there were no differences in morphology, structure, or granule size between *HNF1A* genotypes (Fig. S9d). These results show that *HNF1A* is required for normal dense-core insulin granule formation.

### *HNF1A* deficient scβ-cells are unable to maintain glucose homeostasis in diabetic mice.
To test the ability of *HNF1A* KO β-cells to maintain in vivo normoglycemia, endogenous murine β-cells were ablated by streptozotocin (STZ) 22 weeks post-transplantation. By virtue of the species-related differences in glucose transporters, STZ specifically targets murine β-cells without affecting human β-cells[40]. We first confirmed that STZ treatment did not alter the secretion of c-peptide in sc-β-like cells. There was no change in C-peptide secretion (Fig. S10a) between genotypes in vitro. In vivo, circulating plasma human C-peptide from mice transplanted with *HNF1A* WT or *HNF1A* KO cells was unchanged or increased post STZ treatment (Fig. 5g). Mouse C-peptide was undetectable by ELISA (Fig. S10b) and immuno-histochemistry of the pancreas (Fig. S10c) demonstrating successful ablation of mouse β-cells. Mice transplanted with the *HNF1A* WT β-cells were normoglycemic (~100 mg/dl and HbA1C ~5%) for at least 10 weeks post-STZ injection (Figs. 5h and S10d) and had normal human insulin and C-peptide secretion profiles during an iPGTT (Fig. S10e–g). In contrast, mice transplanted with *HNF1A* KO β-cells did not increase human C-peptide secretion (Fig. 5g) after becoming severely diabetic (blood glucose >500 mg/dl) within one week following STZ injection (Fig. 5h). *HNF1A* KO β-cells were unresponsive to blood glucose even at high levels, failing to maintain systemic glucose homeostasis in a mouse model of β-cell deficient diabetes.

### *HNF1A* R200Q homozygous mutation is pathogenic and causes a developmental bias towards the α-cell fate in vitro.
HNF1A-MODY patients are heterozygous for the causal *HNF1A* alleles. To more directly characterize the consequences of heterozygous patient-specific mutations in *HNF1A*, we differentiated two heterozygous HNF1A-MODY iPSC lines (+/460_461insCGGCATCCAGCACCTGC, ID1056 and +/R200Q), isogenic mutation-corrected HNF1A-MODY iPSC lines R200Q-corrected WT(+/+) and a hESC *HNF1A* Het line (Fig. S2J, C30) into endocrine cells. Heterozygous mutations in *HNF1A* did not affect the generation of definitive endoderm cells (SOX17$^+$) or pancreatic progenitor cells (PDX1$^+$/NKX6.1$^+$) compared to WT lines as determined by immunohistochemistry (Fig. S11a). At the endocrine stage (day 27), organoids were morphologically indistinguishable and all heterozygous cell lines differentiated efficiently to all pancreatic endocrine cells (scα-, scβ- and scδ-like cells); scβ-like cells co-expressed PDX1/NKX6.1/HNF1A (Fig. S11b). HNF1A protein was detected in WT, HNF1A-MODY R200Q Het (+/R200Q), and HNF1A-MODY R200Q-corrected WT (+/+) iPSC-derived endocrine cells by Western blot (Fig. S11c). In contrast to *HNF1A* truncated lines, the heterozygous point mutation R200Q of HNF1A-MODY iPSC line did not affect the total amount of HNF1A protein, suggesting that the mutant R200Q protein was produced (Fig. S11c). No significant differences were found in the proportion of endocrine cell types (scα-, scβ- and scδ-like cells) between *HNF1A* WT and heterozygous cell lines at day 27 of differentiation by immunohistochemistry (Fig. S11d).

To determine HNF1A-MODY patient lines for the bias towards α-cell fate, we measured the percentage of glucagon cells co-expressing C-peptide in two HNF1A-MODY iPSC Het (+/460ins and +/R200Q) lines in vitro. We found no significant differences between HNF1A-MODY iPSC lines and control iPSC R200Q-corrected WT lines (Fig. S11e). To determine the pathogenicity of the *HNF1A* R200Q mutation, we knocked out the *HNF1A* wild type allele in the HNF1A-MODY iPSC Het (+/R200Q) line to make an HNF1A-MODY iPSC Hom (R200Q/−) line, which resulted in a significant ~70% increase in GCG and CPEP co-expressing cells compared to the R200Q-corrected WT control lines (Fig. S11e, f). We also detected a significant increase in the percentage of GCG and CPEP co-expressing cells in hESC *HNF1A* R200Q homozygous (R200Q/R200Q) and hESC *HNF1A* Het (+/−) lines compared to hESC WT (+/+) lines (Figs. S11g and S11h). Thus, while both heterozygous HNF1A-MODY patient cells (+/460ins and +/R200Q) show no developmental bias towards the α-cell fate, complete loss of *HNF1A* WT allele in HNF1A-MODY iPSC R200Q homozygous (R200Q/−) and hESC *HNF1A* R200Q homozygous (R200Q/R200Q) results in a developmental bias towards α-cell fate, indicating the pathogenicity of the R200Q mutation. Interestingly, the hESC heterozygous line harboring a frameshift mutation (Fig. S2J, clone C30) in the DNA-binding domain and causing a premature stop codon presented a strong bias towards α-cell fate (Fig. S11h). Previous studies have shown that frameshift mutations in the DNA-binding domain can cause dominant-negative effects[41].

### *HNF1A* haploinsufficiency gradually impairs scβ-cell functional capacity in the context of metabolic stress.
To assess the developmental potential of HNF1A-MODY patient-specific mutations in vivo, we transplanted pancreatic islet-like clusters derived from iPSCs. While 87.9% (51/58) of mice transplanted with isogenic HNF1A-MODY iPSC Het (+/R200Q) and R200Q-corrected WT (+/+) lines were teratoma-free, the majority (61.5%, 8/13) of the HNF1A-MODY iPSC Het (+/460ins) and control iPSC WT lines showed teratoma formation (Fig. S11i).

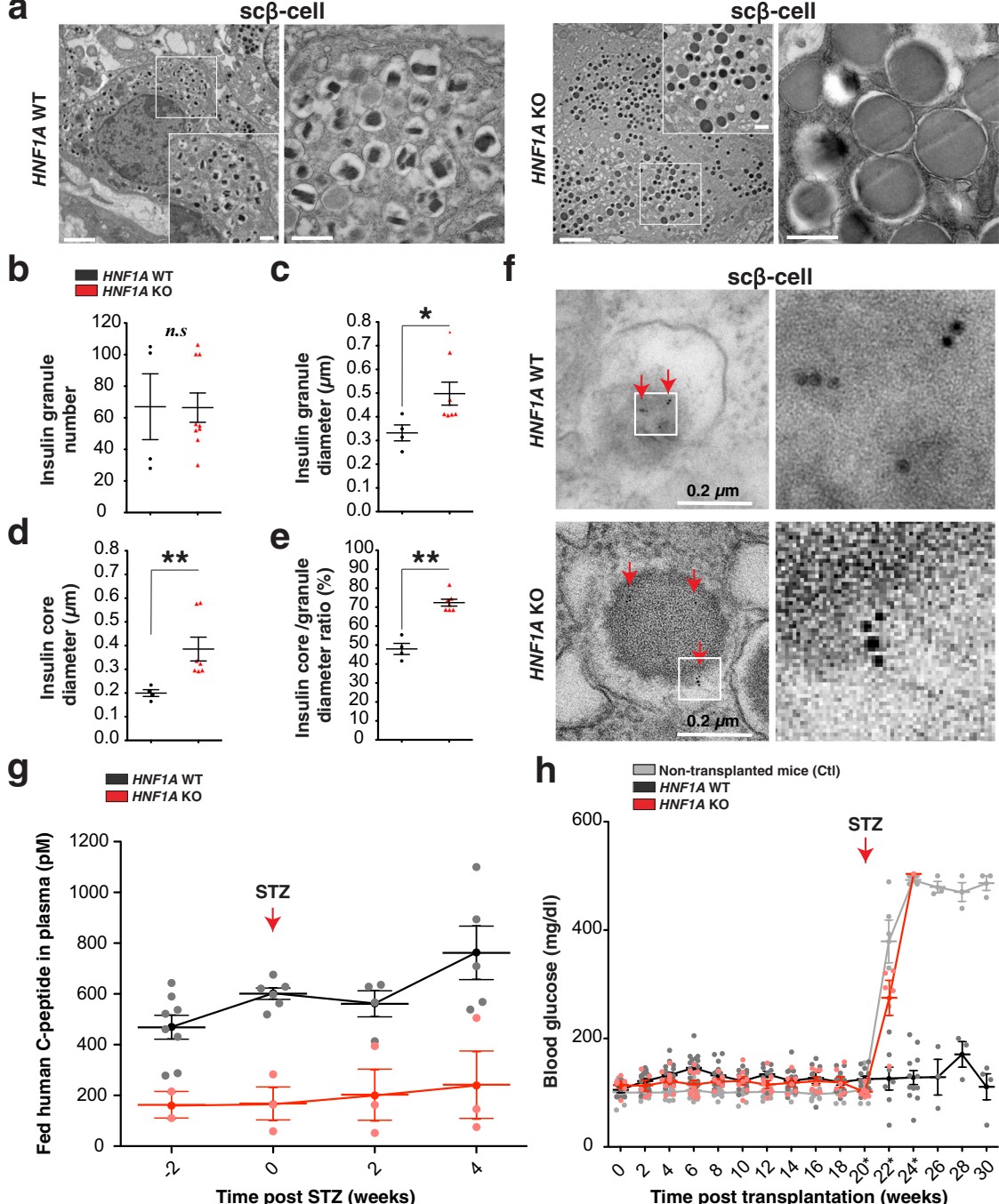

**Fig. 5 HNF1A deficiency causes abnormal insulin granule structure and fails to maintain glucose homeostasis in diabetic mice. a–f** Electron microscopy (EMC) analysis of isolated grafts 30 weeks post-transplantation for hESC-derived endocrine cells (*HNF1A* WT and KO). Explants are from euglycemic mice. **a** Representative EMC images of scβ-cells with **b** quantification of insulin granule number, **c** insulin granule diameter (μm), **d** insulin granule core diameter (μm), and **e** insulin granule core diameter to insulin granule diameter ratio (%) per cell. Each point in the plot is the average of insulin granules per scβ-like-cells. Scale bars: 2 μm in low and 0.5 μm in high magnification. **f** Representative EMC images of scβ-cells with human C-peptide immunogold labeling. Scale bar: 0.2 μm. **g** Fed human C-peptide (pM) in plasma of ad libitum-fed mice transplanted with hESC-derived endocrine cells (*HNF1A* WT *n* = 8 and KO *n* = 3) measured weeks before (−2) and after STZ treatment (0, 2 and 4). **h** Blood glucose concentrations (mg/dl) in ad libitum-fed mice transplanted with hESC-derived endocrine cells (*HNF1A* WT *n* = 18 and KO *n* = 10) or without cells (Ctl *n* = 14). Mice (*HNF1A* WT *n* = 6, KO *n* = 3 and Ctl *n* = 5) were injected (red arrow) with streptozotocin (STZ). All mice were transplanted with hESC-derived endocrine cells at day 27 of differentiation, and grafts were isolated 30 weeks post-transplantation for ex vivo analysis. All protein concentrations were measured by ELISA. (*n*) represents the number of biological replicates. For scatter plots, each point in the plot represents an independent biological experiment (*n*). Data are represented as mean ± SEM. *p*-values: *$p < 0.05$, **$p < 0.01$, ***$p < 0.001$; Mann–Whitney test. n.s. non-significant. See also Figs. S9 and S10.

Because of variable teratoma formation among different iPSC lines[20], only teratoma-free mice were used for further analysis. Teratoma-free transplanted mice had 67.3% (35/52) engraftment efficiency. Thirty weeks post-transplantation, we isolated grafts from normoglycemic animals (Fig. S11j) and found no differences in glucagon-to-insulin content ratios or endocrine cell types between HNF1A-MODY iPSC Het (+/R200Q) and HNF1A-MODY iPSC R200Q-corrected WT (+/+) control grafts (Fig. S11k, l). A recent study of cadaveric HNF1A-MODY human islets with a T260M heterozygous mutation in the DNA-binding domain[29], showed a bias of endocrine cells toward the α-cell fate (Fig. 2k). A bias was also observed in hESC HNF1A heterozygous line with a frameshift mutation in the DNA-binding domain (Fig. S11h). Mutations affecting the DNA-binding domain or truncating mutations are associated with the earlier onset of diabetes in HNF1A-MODY patients[42]. Thus, different mutations affect HNF1A function to different degrees, resulting in phenotypic differences among patients as well as characteristic cellular phenotypes in a stem cell system.

To understand the function of HNF1A-MODY iPSC Het β-cells in a physiological context, we monitored circulating human insulin and C-peptide in transplanted euglycemic mice over 30 weeks. Mice transplanted with iPSC-derived cells from heterozygous HNF1A-MODY patients (+/460_461insCGGCAT CCAGCACCTGC) had low, but detectable circulating human C-peptide 2, 4, 8 (Fig. S12a) and 30 weeks post-transplantation (Fig. S12b) with undetectable human insulin (Fig. S12c). Since human insulin levels are under the minimal threshold limit for detection by ELISA, this deficiency can be attributed to the lower ability of these mutant iPSCs to generate functional insulin-producing cells in vivo. In mice transplanted with iPSC-derived cells from HNF1A-MODY patient (+/R200Q) and R200Q-corrected WT (+/+) isogenic lines, circulating human C-peptide (Fig. S12b) and human insulin (Fig. S12c) concentrations reached comparable levels 30 weeks post-transplantation. Transplantation of mice with hESC HNF1A Het (+/−), thought to be a dominant negative allele, and hESC HNF1A R200Q homozygous (R200Q/R200Q) line resulted in low circulating concentrations of human C-peptide over 30 weeks post-transplantation (Fig. S12a, b) accompanied by low to undetectable human insulin (Fig. S12c). These results show that HNF1A heterozygous frameshift mutations in the DNA-binding domain (hESC +/−) and HNF1A R200Q homozygosity (hESC R200Q/R200Q) are pathogenic and that heterozygous patient mutations (iPSC +/R200Q) are less detrimental to β-cell function.

To further interrogate the consequence of the R200Q mutation in β-cells, we performed intraperitoneal glucose tolerance tests (iPGTT) in mice transplanted with the HNF1A-MODY iPSC Het (+/R200Q) and R200Q-corrected WT (+/+) isogenic lines, as these iPSC lines allowed the reliable generation of transplanted mice with high C-peptide and insulin levels over 30 weeks post-transplantation (Fig. S12d, e). During an iPGTT, mice transplanted with both isogenic lines showed comparable changes in blood glucose (Fig. S12f) and plasma human C-peptide (Fig. 6a); plasma human insulin concentrations dropped below detection during fasting and increased upon glucose injection (Fig. 6b), demonstrating homeostatic glucose responsiveness. Glibenclamide injection increased C-peptide and insulin concentrations in both genotypes (Fig. S12g, h) at equivalent ratios (Fig. S12i). No significant differences in insulin secretion (Fig. S12j) or endocrine hormone content were found in vitro or in vivo between genotypes (Fig. S12k, l).

HNF1A-MODY patients generally display normal glucose tolerance during early childhood and exhibit symptomatic diabetes in their late teens or early adulthood[9] depending on the type and position of the HNF1A mutation[42]. To determine whether there are

functional differences between HNF1A-MODY heterozygous (+/R200Q) and HNF1A-MODY R200Q-corrected WT (+/+) scβ-cells, we treated mice with STZ, thereby exposing transplanted scβ-cells to higher insulin demand due to ablation of mouse endogenous β-cells. Two weeks post STZ, animals transplanted with HNF1A-MODY iPSC R200Q Het (+/R200Q) cells showed rapid progression to hyperglycemia (Fig. 6c). The increase in blood glucose (mice n = 9) was accompanied by a failure to increase circulating human C-peptide (Fig. 6d) and a reduction of human insulin concentrations (Fig. 6e) over 5–6 weeks after STZ treatment. In contrast, in mice (n = 6) transplanted with isogenic HNF1A-MODY iPSC R200Q-corrected WT cells (+/+), concentrations of human plasma C-peptide (Fig. 6d) and insulin increased during this period (Fig. 6e).

Five to six weeks post STZ treatment, we found that the ratio of circulating insulin to C-peptide fell by 55% in mice transplanted with HNF1A-MODY iPSC R200Q Het (+/R200Q) cells. In contrast, insulin:C-peptide ratios remained constant in mice transplanted with HNF1A-MODY R200Q-corrected WT (+/+) cells (Fig. 6f). The decrease in circulating insulin:C-peptide ratios were reciprocal to a significant increase by 74% in the insulin:C-peptide ratios of intracellular content from isolated HNF1A-mutated grafts compared to isogenic controls (Fig. 6g), demonstrating complementary imbalance of insulin to C-peptide secreted. These results show that HNF1A R200Q haploinsufficiency gradually impairs the stoichiometry of circulating insulin:C-peptide, and that gene correction of the R200Q mutation protects HNF1A-MODY iPSC β-cells from acquiring this imbalance.

In sequential iPGTTs 2–8 weeks post-STZ administration, HNF1A-MODY iPSC R200Q Het (+/R200Q) scβ-cells showed progressive impairment of glucose-stimulated insulin secretion: mild at two weeks post-STZ (Fig. S13a, b) and completely unresponsive to glucose challenges four weeks post-STZ (Fig. S13c, d). Mice transplanted with HNF1A-MODY scβ-cells fail to clear glucose from the circulation during a glucose tolerance test at 6–8 weeks post STZ treatment (Figs. 6h and S13e). HNF1A R200Q heterozygous (+/R200Q) scβ-cells showed no glucose-stimulated insulin secretion, whereas HNF1A-MODY R200Q-corrected WT (+/+) scβ-cells were sensitive to fasting, remained glucose responsive 6–8 weeks post STZ treatment, and cleared glucose from circulation (Figs. 6h, i and S13e, f). Mice transplanted with HNF1A-MODY iPSC R200Q Het (+/R200Q) scβ-cells became diabetic as reflected by elevated blood glucose (Fig. S13g) and HbA1c levels (Fig. 6j) compared to control mice. These results show that HNF1A-MODY iPSC (+/R200Q) β-cells fail to compensate for higher metabolic insulin demands and that hyperglycemia is due to the gradual development of insulin secretory defects. Correction of R200Q mutations protects HNF1A-MODY iPSC β-cells from acquiring abnormal insulin secretion profiles.

## Discussion

Here we report the use of stem cell-derived islets containing HNF1A mutations to elucidate the molecular basis for β-cell dysregulation of insulin secretion in HNF1A hESC null and HNF1A-MODY hypomorphic (R200Q) patient lines. In HNF1A knockout stem cell-derived β-cells, glucose-stimulated insulin secretion is compromised in association with a reduction in CACNA1A and intracellular calcium levels, and down-regulation of SYT13, a gene involved in calcium-regulated granule exocytosis. Mutations in these genes reproduce the insulin secretion phenotypes found on the stem cell model of HNF1A deficiency. These differences in gene expression were seen in both knockout and hESC-derived cells rendered homozygous for R200Q

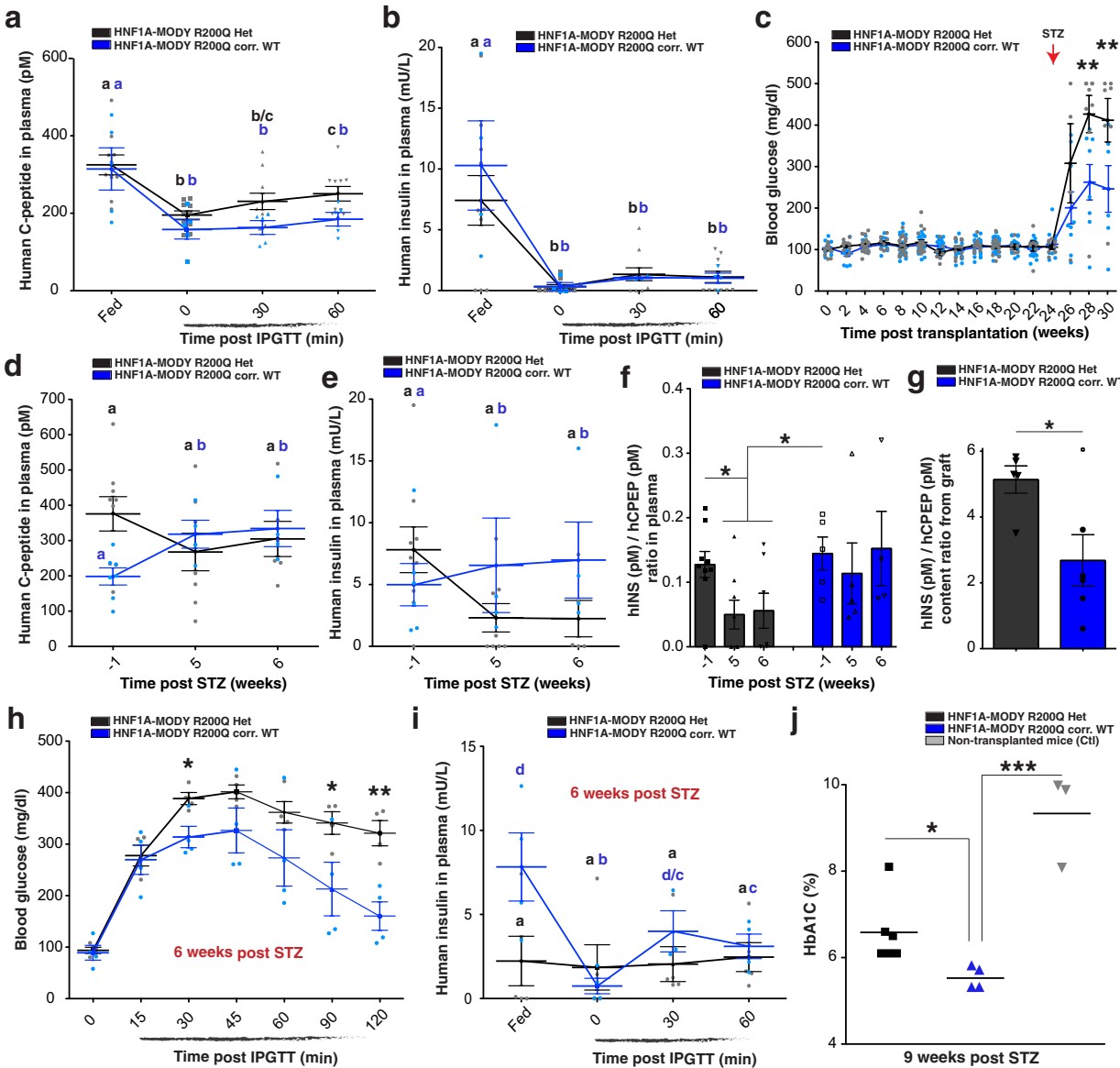

**Fig. 6 HNF1A haploinsufficiency gradually impairs scβ-cell function in vivo. a–c** Human C-peptide (pM) and human insulin levels in plasma of ad libitum-fed mice transplanted with HNF1A-MODY iPSC-derived endocrine cells (*HNF1A* +/R200Q and R200Q-corrected WT). **a** Plasma human C-peptide (pM) and **b** human insulin secretion (mU/L) in mice transplanted with HNF1A-MODY iPSC-derived endocrine cells (R200Q Het $n = 10$ and R200Q corr. WT $n = 5$) in ad libitum-fed state and during an iPGTT (t0, t30, and t60). *p*-values were **b**: $p < 0.001$, **b**, **c**: $p < 0.01$ and **c**: $p < 0.05$. **c** Blood glucose concentrations (mg/dl) in ad libitum-fed mice transplanted with HNF1A-MODY iPSC-derived endocrine cells (R200Q Het $n = 21$ and R200Q corr. WT $n = 32$). Mice (R200Q Het $n = 8$, R200Q corr. WT $n = 8$ and Ctl $n = 5$) were injected (red arrow) with STZ. **d** Human C-peptide (pM), **e** human insulin (mU/L) (R200Q Het $n = 9$ and R200Q corr. WT $n = 6$), and **f** hINS (pM)/hCPEP (pM) secretion ratios monitored in plasma of ad libitum-fed mice weeks before (−1) and after (5 and 6) STZ treatment. *p*-values were **b**, **d**: $p < 0.01$; **c**: $p < 0.05$. **g** Quantification of hINS(pM)/hCPEP(pM) content ratios from isolated grafts. **h**, **i** IPGTT in mice transplanted with HNF1A-MODY iPSC-derived endocrine cells (R200Q Het $n = 5$ and R200Q corr. WT $n = 4$) 6 weeks post STZ treatment in ad libitum-fed state and during an iPGTT (t0, t30 and t60). **h** Blood glucose concentrations (mg/dl) and **i** human insulin secretion (mU/L) in plasma. *p*-values were **b** $p < 0.01$, **d/c** and **c**: $p < 0.05$. (R200Q Het $n = 5$ and R200Q corr. WT $n = 4$). **j** HbA1C (%) in mice transplanted with HNF1A-MODY -iPSC-derived endocrine cells or non-transplanted (Ctl) 9 weeks after STZ treatment. All mice were transplanted with iPSC-derived endocrine cells at day 27 of differentiation, and grafts were isolated 30–35 weeks post-transplantation for ex vivo analysis. All protein concentrations were measured by ELISA. For scatter plots, each point in the plot represents an independent biological experiment (*n*). Data are represented as mean ± SEM. Different letters designate significant differences within groups. *p*-values were *$p < 0.05$, **$p < 0.01$, ***$p < 0.001$; two-tailed *t*-test. n.s. non-significant. See also Figs. S11–S13.

(R200Q/R200Q), confirming the pathogenicity of the R200Q allele. The deficiency in insulin secretion can be overcome through the application of sulfonylureas, by increasing the low calcium levels found in *HNF1A* mutant cells. Thereby we demonstrate the molecular basis for the efficacy of sulfonylureas

in the treatment of HNF1A-MODY. However, HNF1A deficiency has pleiotropic effects, which will need to be further elucidated for effective treatment long term. The triggering of insulin release through sulfonylureas often becomes insufficient at later stages of the disease.

In particular, HNF1A also has developmental functions; hypomorphic mutations in *HNF1A* perturb the expression of genes important in β-cell identity such as *PAX4* and lead to increases in glucagon gene expression, α-cell number, and the α- to β-cell ratio in mature pancreatic islets. This α-cell bias phenotype was observed in homozygous hESC *HNF1A* knockouts as well as homozygous hESC and iPSC for R200Q (iPSC R200Q/− and in hESC R200Q/R200Q) but was not seen in HNF1A-MODY patient iPSC-derived heterozygous cells (+/460ins and +/R200Q). A recent case study of cadaveric human islets from a 33-year-old HNF1A-MODY donor with a heterozygous mutation (+/T260M) in the *HNF1A* DNA-binding domain and with a 17 years history of diabetes also showed an increase in the ratio of α- to β-cells[29].

The accumulation of abnormal insulin secretory granules in *HNF1A* knockout β-cells was associated with a deviation of the 1:1 stoichiometric release of C-peptide and insulin from β-cells, characterized by the constitutive release of C-peptide and intracellular retention of insulin. Their equimolar secretion was restored by depolarization of the cells with sulfonylurea. Previous studies have identified a "constitutive-like" secretory pathway that is characterized by the accumulation of immature secretory granules that secrete newly synthesized C-peptide in molar excess of insulin[43,44]. Our data are consistent with this form of secretion, but a detailed study of insulin secretory dynamics in HNF1A-MODY patients is needed to fully elucidate the clinical contributions of the secretory defects described here. We cannot formally exclude the possibility that insulin secreted from mutant cells is more efficiently removed from the circulation than insulin secreted from wild type cells.

The insulin secretory phenotype of heterozygous HNF1A-MODY iPSC-derived β-cells differed from complete *HNF1A* deficiency. Heterozygous β-cells (+/R200Q) initially functioned normally in transplanted mice, consistent with normal glucose tolerance of heterozygous carriers during early childhood, followed by symptomatic diabetes within the first three decades of life (Supplementary Data 6 for the age of onset)[9]. In response to increasing the requirements for human insulin following ablation of the mouse's intrinsic β-cells, iPSC-derived β-cells from HNF1A-MODY patients were unable to fully compensate for the increased insulin requirement, and gradually developed phenotypes resembling those of *HNF1A* knockout β-cells, including a failure to secrete insulin in response to glucose, and reduced insulin to C-peptide secretion ratios. Correction of the R200Q mutation in HNF1A-MODY iPSC-derived β-cells protected cells from acquiring abnormal insulin secretion profiles.

Our study illustrates both congruences as well as differences with other stem cell models of HNF1A deficiency. Our data and another study identify a bias in endocrine differentiation to glucagon-positive cells in *HNF1A* knockout cells[45]. Cardenas-Diaz also documents this bias in heterozygous cells, to an extent comparable to biallelic knockout cells. Mutations in *HNF1A* introduced by Cardenas-Diaz are located adjacent to the start codon, excluding possible dominant-negative effects of a truncated protein. Such dominant-negative truncating mutations were made by another group and shown to affect pancreatic differentiation from pluripotent stem cells by interfering with *HNF1B* function[46]. Both our study and Cardenas-Diaz et al. identify a deficiency of glucose-stimulated insulin secretion in *HNF1A* biallelic knockout cells[45]. Surprisingly, Cardenas-Diaz and colleagues found that heterozygous and homozygous knockout cells were equally impaired in insulin secretion. The finding that haploinsufficiency is as detrimental as a complete knockout or a dominant negative mutation across different phenotypic aspects, is surprising and was not observed here. Cardenas-Diaz et al. linked insulin secretion phenotypes to altered mitochondrial function decreased expression of *LINC01139* and increased expression of *LDHA* involved in anaerobic glycolysis, reporting similar changes in expression of these genes in both heterozygous and homozygous mutant cells. An increase of *LDHA* due to *HNF1A* deficiency was not observed in our study, neither in bulk-RNAseq nor in single-cell RNAseq, and neither in *HNF1A* knockout or heterozygous cells. Neither did Haliyur et al. see these changes in the pancreatic islets of an HNF1A-MODY heterozygous (+/T260M) donor[29]. *LDHA* is highly expressed in undifferentiated pluripotent stem cells; its expression may thus be affected by the maturity of stem cell-derived pancreatic cell clusters. An intriguing concordance of phenotypes of a stem cell model (Cardenas-Diaz et al.) and islets from an HNF1A-MODY patient (Haliyur et al.) is the increased basal secretion rate of insulin. However, such phenotypic concordances among studies are not necessarily due to concordant mechanisms. Other factors, including increased *LDHA* expression and rates of glycolysis, may also affect basal insulin secretion in stem cell-derived insulin-producing cells. Interestingly, we find that *HNF1A* knockout cells show reduced expression of *KCNH6*, mutations which are associated with elevated insulin secretion[47]. Another study has found downregulation of glucose transporter *GLUT2* expression and reduced ATP generation in HNF1A-MODY patient iPS-derived β-cells with heterozygous mutation (+/H126D) in comparison to control wild type lines of a different genetic background[48]. In summary, these stem cell-based studies on HNF1A deficiency reveal pleiotropic effects of *HNF1A* deficiency at several levels of β-cell biology and function, consistent with the diversity of its transcriptional targets.

Among the clinical implications of this study are that stem cell-derived human islet cell clusters capable of supporting glucose homeostasis in mice. In MODY patients, β-cell autoimmunity is not mechanistically involved, so islets created from somatic cells of such patients should be immunologically tolerated by the patient. As shown here, the mutant alleles of HNF1A-MODY patients could be corrected to produce autologous stem-cell-derived islets with a normal *HNF1A* allele. While sulfonylureas can be used to treat patients with HNF1A-MODY effectively, insulin dependence is common after years of treatment. For these patients, cell therapy might be considered.

## Methods

**Human subject and cell lines**. Nine HNF1A-MODY subjects (Pt1–Pt9), three MODY2 subjects (1068, 1133 and 1144), and four control subjects (1023, 1098, 1136 and 1015) were recruited at the Naomi Berrie Diabetes Center and monitored over 2–5 years. Samples were coded to protect subjects' identities (Fig. S2j). Biopsies from HNF1A-MODY subjects (Pt1, Pt2 and Pt3) and one control subject (1023) were cut into small pieces (approximately $5 \times 5$ mm in size). 2–3 pieces of minced skin were placed next to a droplet of silicon in a well of the six-well dish. A glass coverslip ($22 \times 22$ mm) was placed over the biopsy pieces and silicon droplet. 5 ml of biopsy plating media was added and incubated for 5 days at 37 °C. Biopsy pieces were then grown in a culture medium for 3–4 weeks. Biopsy plating medium is composed of DMEM (Gibco, 10569), 10% FBS (GE Healthcare, SH30088.03HI), 1% GlutaMAX (Gibco, 35050061), 1% Anti–Anti (Gibco, 15240062), NEAA (Gibco, 11140-050), 0.1% 2-Mercaptoethanol (Gibco, 21985023) and nucleosides (Millipore ES-008-D). Culture medium contains DMEM (Gibco, 10569), 10% FBS (GE Healthcare, SH30088.03HI), 1% GlutaMAX (Gibco, 35050061) and 1% Pen-Strep (Gibco, 15070063). The hESCs (Mel1)[21] used in this manuscript is an NIH-approved line. All research involving human subjects was approved by the Institutional Review Board (IRB) of Columbia University Medical Center, and all participants provided written informed consent.

**Generation of iPSCs and cytogenic analysis of stem cells**. Primary fibroblasts were reprogrammed into pluripotent stem cells using CytoTune™-iPS Sendai Reprogramming Kit (Invitrogen). 50,000 fibroblast cells (between passages 2 and 5) were seeded in a well of a six-well dish and allowed to recover overnight. The next day, cells were infected by the Sendai virus expressing human transcription factors Oct4, Sox2, Klf4, and C-Myc mixed in fibroblast medium according to the manufacturer's instructions. Two days later, the medium was exchanged for human ES medium supplemented with the ALK5 inhibitor SB431542 (2 μM; Stemgent), the MEK inhibitor PD0325901 (0.5 μM; Stemgent), and thiazovivin (0.5 μM; Stemgent). Human ES medium contained KO-DMEM (Gibco 10829), 15% KnockOut Serum Replacement (Gibco, 10828), 1% GlutMAX, 0.1% 2-Mercaptoethanol, 1%

NEAA, 1% PenStrep and 0.1 µg/mL bFGF (all from Gibco). On days 7–10 post-infection, cells were detached using TrypLE™ Express (Gibco, 12605036) and passaged onto mouse embryonic fibroblast feeder cells (GlobalStem CF-1 MEF IRR). Individual colonies of induced pluripotent stem cells were manually picked between day 21 and 28 post-infection and each iPSC line was expanded from a single colony. All stem cell lines were cultured on a human ES medium. Cytogenic analysis was performed on 20 G-banded metaphase cells from each line by Cell Line Genetics Inc (Fig. S2m).

**Stem cell culture**. hESC and iPSCs were grown on plates coated with primary mouse embryonic fibroblasts or MEFs (GlobalStem, CF-1 MEF IRR) and dissociated every 4–5 days using TrypLE™ Express (Life Technology, 12605036) for passaging. After dissociation, cells were suspended in a human ES medium containing 10 µM ROCK inhibitor Y27632 (Selleckchem, S1049).

**In silico gRNA design**. sgRNAs (Supplementary Data 1) were designed using an online CRISPR design tool (crispr.mit.edu) and cloned into a gRNA cloning vector (Addgene, 41824) following option B from the gRNA Synthesis Protocol[49]. The resulting vector was transformed into competent bacteria using Gibson Assembly® chemical transformation protocol (E5510). Single clones were picked and grown on 3 ml of LB broth for 16–18 h at 37 °C in a shaker (250 rpm); DNA is extracted and Sanger sequenced.

**CRISPR/Cas9 nucleofection and mutagenesis**. Stem cells were cultured with human ES media containing ROCK inhibitor Y27632 (Selleckchem, S1049) 3 h prior to nucleofection, dissociated, and filtered through a 70 µm cell strainer (Thermo Fisher Scientific, 8-771-2). Approximately $2 \times 10^6$ cells were nucleofected (Lonza nucleofector, program A23) with 5 µg of Cas9-GFP plasmid (Addgene, 44719), 5 µg of sgRNA and 5 µg of ssDNA donor template (Supplementary Data 1) using Human Stem Cell Nucleofector™ kit 1 (Lonza, VVPH-5012) according to the manufactures protocol and cells replated. Cells were dissociated 48 h later for GFP sorting using BD FACS Aria II cell sorter. As a quality control step, some unsorted cells were used to test the sgRNAs mutation efficiency using the Transgenomic SURVEYOR® mutation detection kit according to the manufacturer's protocol. Sorted cells were plated in a 10 cm dish and single clones were picked 7–10 days post sorting, clones were further expanded and DNA was extracted for PCR using HNF1A primers (Supplementary Data 1). Amplicons were sent for Sanger sequencing to GENEWIZ and clones with indels were further validated by TOPO® TA cloning (Thermo Scientific, 450641) (at least 6 clones were picked) followed by Sanger sequencing. For the hESC WT line, five different clonal lines were used for analysis throughout the study.

**Differentiation into pancreatic endocrine cells**. Cells were grown to 80–90% confluency, dissociated, and suspended in mTeSR™ medium (STEMCELL Technology, 05850) with 10 µM ROCK inhibitor Y27632 (Selleckchem, S1049) and plated in a 1:1 ratio into Matrigel-coated (Fisher Scientific, 354277) wells for differentiation. Differentiation was performed using a published protocol[20]. The initial stages of differentiation were conducted in planar culture (d0–d11). For the definitive endoderm stage (d1–d3) cells were cultured using STEMdiff™ Definitive Endoderm Differentiation Kit (Stemcell Technologies, 05110). For primitive gut stage (d4–d6), cells were cultured in RPMI containing GlutaMAX (Life Technology, 61870-127), 1% (v/v) Penicillin–Streptomycin (PS) (Thermo Fisher Scientific, 15070-063), 1% (v/v) B27 Serum-Free Supplement (50×) (Life Technology, 17504044) and 50 ng/ml FGF7 (R&D System, 251-KG). For posterior foregut stage (d7 and d8), cells were cultured in DMEM -containing GlutaMax, 1% (v/v) PS, 1% (v/v) B27, 0.25 µM KAAD-Cyclopamine (Stemgent, 04-0028), 2 µM Retinoic acid (Stemgent, 04-0021) and 0.25 µM LDN193189 (Stemgent, 04-0074). For pancreatic progenitor stage (d9–d11), cells were cultured in DMEM containing GlutaMax, 1% (v/v) PS, 1% (v/v) B27, and 50 ng/ml EGF (R&D System, 236-EG). Cells were then dissociated using TrypLE™ Express (Life Technology, 12605036) and seeded into low-attachment 96 well-plates (Corning, 7007) (1 well of six-well-plate to 60 wells of 96-well-plate) for clustering step to form aggregates or clusters of endocrine cells in DMEM containing GlutaMax, 1% (v/v) PS, 1% (v/v) B27, 0.25 µM Cyclopamine, 1 µM thyroid hormone (T3) (Sigma, T6397), 10 µM Alk5i, 10 µM Zinc sulfate (Sigma-Aldrich, Z4750) and 10 µg/ml Heparin (Sigma-Aldrich, H3149) for 2 days (d12–d13). For pancreatic endocrine stage (d14–d20) cells were cultured using DMEM containing GlutaMax, 1% (v/v) PS, 1% (v/v) B27, 100 nM LDN, 1 µM T3, 10 µM Alk5i, 10 µM Zinc sulfate, 10 µg/ml Heparin and 100 nM gamma-secretase inhibitor (DBZ) (EMD Millipore, 565789). For mature pancreatic endocrine stage (d21–d27) cells were cultured using DMEM-containing GlutaMax, 1% (v/v) PS, 1% (v/v) B27, 1 µM T3, 10 µM Alk5i, 10 µM Zinc sulfate, 10 µg/ml Heparin, 1 mM N-acetyl cysteine (N-Cys) (Sigma-Aldrich, A9165-5G), 10 µM Trolox (EMD Millipore, 648471-500MG) and 2 µM R428 (Tyrosine kinase receptor AXL inhibitor) (ApexBio, A8329). From d1 to d11 media was changed every day and from d12 to d27 media was changed every other day. All differentiations were done for 27 to 30 days.

**In vitro insulin secretion and content**. Static insulin secretion assay was performed on days 27–30 of differentiation; 10 islet-like clusters of cells were used per experiment. Islet-like clusters were pre-incubated for 1 h in Kreb's Ringer Buffer (128 mM NaCl, 5 mM KCl, 2.7 mM CaCl₂, 1.2 mM MgSO₄, 1 mM NaHPO₄, 1.2 mM KH₂PO₄, 5 mM NaHCO₃, 10 mM HEPES, 0.1% Bovine Serum Albumin, pH = 7.4) containing 3.3 mM glucose, washed and incubated for another hour in 3.3 mM glucose and the medium was collected. Subsequently, 200 µl of buffer containing 16.7 mM glucose or 400 µM tolbutamide (abcam, ab120278) or 30.5 mM KCl was used to treat cells for 1 h, after which the medium was collected. Insulin content was measured by acid ethanol extraction; cells were resuspended in 50 µl of water and sonicated for 15 s. The sonicate is mixed with acid ethanol (0.18 M HCl in 96% ethanol (vol/vol)), in a 1:3 ratio of sonicate to acid ethanol. The mixed solution is incubated at 4 °C for 12 h. Human C-peptide, human insulin, and human proinsulin secretion and content were measured using Ultrasensitive C-peptide ELISA (Mercodia, 10-1141-01), Insulin ELISA (Mercodia, 10-1113-01), and Proinsulin ELISA (Mercodia, 10-1118-01) kit according to the manufacturer's protocol. Mouse insulin was measured using an Ultrasensitive Insulin ELISA (Mercodia, 10-1249-01) kit according to the manufacturer's protocol. All samples were handled the same way.

**Dynamic insulin secretion**. Perifusion was performed as follows and according to ref. [50]. Twenty randomly chosen clusters of cells were examined using a Biorep Technologies (Miami, FL) perifusion system. Clusters were perifused with Kreb's buffer [115 mM NaCl, 5 mM KCl, 24 mM NaHCO₃, 1 mM MgCl₂, 2.2 mM CaCl₂ at pH 7.4] supplemented with 0.17% bovine serum albumin and 3.3 mM glucose (26 min), followed by 16.7 mM glucose (35 min), 3.3 mM glucose (15 min), 400 µM tolbutamide in 3.3 mM glucose (35 min), 20 mM KCl plus 3.3 mM glucose (10 min) and finally 3.3 mM glucose (15 min). The medium was collected at a flow rate of 100 µl/min to assess insulin secretion. Insulin concentration was measured using an insulin ELISA kit (Alpco, 80-INSHU-E01.1). Clusters of cells were collected at the end of the study and placed in acidified ethanol overnight to determine total insulin levels. All samples were handled the same way.

**Calcium Imaging**. Stem cell-derived β-cells were dissociated from clusters and plated onto 35 mm glass-bottom dishes coated with 5% Matrigel (Fisher Scientific, 354277). Cells were washed twice with basal KRBH solution composed of (mM): 128 NaCl, 5 KCl, 2.7 CaCl₂, 1.2 MgSO₄, 1 NaHPO₄, 1.2 KH₂PO₄, 5 NaHCO₃, 10 HEPES, 0.1% Bovine Serum Albumin, 3.3 glucose, pH to 7.4. Cells were then incubated in the same solution containing 1 µM fura-2, AM (Thermo Fischer, F1221) with 0.05% pluronic F-127 in DMSO (Thermo Fischer, P3000MP) for 15 min at 37 °C, 5% CO₂. Cells were washed twice with basal KRBH solution and then imaged on an inverted Nikon Ti-eclipse microscope with a Nikon Plan fluor ×20 objective (0.45 NA). Fura-2 measurements were collected at excitation wavelengths of 340 and 380 nm using EasyRatioPro (HORIBA Scientific). Stimulation solutions included either 16.7 mM glucose, 600 µM tolbutamide, or 30.5 mM KCl, with NaCl concentrations adjusted accordingly to balance osmolarity with KRBH solution.

Calcium concentrations were calculated as follows:

$$[Ca^{2+}] = K_d \cdot \frac{R - R_{min}}{R_{max} - R} \cdot \frac{S_{f2}}{S_{b2}} \qquad (1)$$

where $K_d$, the apparent Ca²⁺ binding affinity to Fura-2, was assumed to be 225 nM and $R_{max}$, $R_{min}$, $S_{f2}$, and $S_{b2}$ values were obtained using 10 µm ionomycin with either no Ca²⁺ or 2 mM Ca²⁺, values[51,52].

**Immunohistochemistry**. Clusters of cells and tissue (graft or pancreas) were fixed with 4% paraformaldehyde for 20 min at room temperature or 24 h at 4 °C respectively, washed in PBS, then dehydrated with 30% sucrose for 24 h. Cells and tissues were washed in PBS, then cryopreserved in frozen OCT and sectioned at 7 µm on microscope slides. Slides were then incubated with a blocking solution containing 3% donkey serum with 0.1% triton X-100 diluted in PBS for 1 h at room temperature. Primary antibodies (Supplementary Data 7) were added in block solution overnight at 4 °C, washed three times for 5 min in PBST (PBS with 0.1% Tween-20) and incubated with secondary antibodies (Supplementary Data 7) diluted in block solution for 1 h at room temperature, washed three times in PBST containing 10 µg/ml DAPI for 5 min and mounted with a coverslip in fluorescent mounting media (Dako, S3023). All images were taken using a Zeiss LSM 710 confocal microscope and quantified manually using ImageJ software. Quantification of cell type at the definitive and pancreatic progenitor stage (planar culture) was done by randomly choosing 4 sections in the well and averaging the percentage of cells from all sections. For clusters of cells (3D culture) and tissue (graft or pancreas), they were cut in 3–6 sections spaced 20 and 100 µm, respectively, and cell type was quantified in each section as a percentage. The final quantification result is the average of all sections. The ratio of hormone-positive cells for each insulin and glucagon over the total number of cells identified by DAPI nuclear staining was quantified using ImageJ and reported as the fraction of total pancreatic parenchyma for each hormone (i.e. hormone-positive area). Each cluster of cells is comprised of ~10,000 cells.

**Flow cytometry**. Clusters of endocrine cells were dissociated into single cells using TrypLE™ Express (Life Technology, 12605036). Cells were then fixed with 4%

paraformaldehyde for 20 min at room temperature followed by 10 min permeabilization with cold methanol at −20 °C. Cells were washed with 3% donkey serum diluted in PBS and primary antibodies diluted (Supplementary Data 7) in a blocking solution containing 3% donkey serum with 0.1% triton X-100 diluted in PBS overnight at 4 °C. For tunel assay (Biotium, 30074) cells were incubated with fluorescent biotinylated nucleotide conjugate according to the manufacturer's instructions. Cells were washed in PBST (PBS with 0.1% Tween-20) and incubated with secondary antibodies diluted (Supplementary Data 7) in block solution for 1 h at room temperature. Cells were washed and filtered with a BD Falcon 12 × 75 mm tube with a cell strainer cap (BD Biosciences, 352235) and analyzed by flow cytometer BD Fortessa. Data were analyzed using FlowJo software. Gating for flow cytometry was determined by incubating the cells only with secondary antibodies (negative control) and was consistent across experiments (shown in magenta in the figures).

**Western Blot**. Clusters of endocrine cells were dissociated into single cells using TrypLE™ Express (Life Technology, 12605036). Cells were lysed using RIPA buffer (NP40 1%, NaCl 150 mM, EDTA 1 mM, Tris–HCl pH 7.5 50 nM, SDS 0.1%, sodium deoxychol 0.5%, NaF 10 mM, protease inhibitor tablet) and whole-cell lysates obtained by subsequent centrifugation. Immunoblots were incubated with primary antibodies against HNF1A (abcam, ab96777, 1:500) and β-Tubulin-III (Sigma, T2200, 1:500) (Supplementary Data 7). Raw images are provided in Fig. S14.

**Bulk RNA sequencing**. Around 200 clusters of cells were dissociated and sorted for GFP with BD FACS Aria II cell sorter. Sorted cells were pelleted by centrifugation and RNA was extracted using total the RNA purification micro (Norgen Biotek, 35300) kit according to the manufacturer's protocol. RNA quality and concentration are determined by an Agilent bioanalyzer and RNAseq performed by the Columbia University Genome Center. A poly-A pull-down was used to enrich mRNAs from total RNA samples (200 ng–1 μg per sample, RIN > 8 required) and proceed with library preparation by using the Illumina TruSeq RNA prep kit. Libraries were then sequenced using an Illumina HiSeq2000. RTA (Illumina) was used for base calling and bcl2fastq (version 1.8.4) for converting BCL to fastq format, coupled with adaptor trimming. The reads were mapped to a reference genome (Human: NCBI/build37.2; Mouse: UCSC/mm9) using Tophat[53] (version 2.1.0) with 4 mismatches (–read-mismatches = 4) and 10 maximum multiple hits (–max-multihits = 10). To tackle the mapping issue of reads that are from exon-exon junctions, Tophat infers novel exon-exon junctions ab initio and combines them with junctions from known mRNA sequences (refgenes) as the reference annotation. The relative abundance (aka expression level) of genes and splice isoforms is estimated using cufflinks[54] (version 2.0.2) with default settings. Differentially expressed genes were tested under various conditions using DEseq[55]. An R package based on a negative binomial distribution that models the number reads from RNA-seq experiments to test for differential expression.

**Single-cell RNA sequencing**. Around 100 clusters of cells per genotype at day 27 of differentiation were dissociated and sorted for GFP using BD FACS Aria II cell sorter. Single-cell RNA-Seq data from a total of 271 hESC-derived cells (113 for WT and 158 for KO) were obtained. Raw reads were aligned to the hg19 reference genome using STAR (2.5.2b) and gene quantification was performed using FeatureCounts (1.5.2). Quality control was conducted as follows: cells with low mapped reads (<1,000,000), low exon count (<100,000), low exon mapped rate (<0.2), high intergenic mapped rate (>0.3), low detected gene count (<1000) and with more than 10% Mitochondria genes' count were removed. In addition, genes with <10 non-zero counts across all the cells were eliminated. All the downstream analysis was done within R-3.4.1. Filtered raw count data were normalized and scaled by the Seurat package and only variable genes generated by Seurat were used to do PCA and t-SNE. $n = 3$ for each genotype.

**10X genomics single-cell RNA sequencing**. Around 100 clusters of cells per genotype at day 27 of differentiation were dissociated. Single cells were suspended in PBS + 0.04% BSA. Cellular suspensions (~6000 cells) were loaded on a Chromium Single Cell Instrument (10X Genomics) to generate single cell GEMs. Single-cell RNAseq libraries were prepared using Chromium Single cell 3¢ Library, Gel beads & Multiplex kit (10X Genomics). Sequencing was performed on Illumina NextSeq500 using the following read length: 59-bp Read1 for transcript read, 14-bp I7 Index for Cell Barcode read, 8-bp I5 Index for sample index read, and 10-bp Read2 for UMI read. The samples were sequenced in ten different libraries (batches). Among all these samples, Batch 980379 and Batch 993091 (2 libraries) are HNF1A mutants (R200Q/R200Q), Batch 980380 and Batch 993090 (2 libraries) are HNF1A knockouts (KO) while Batch 989516 and Batch 993089 (2 libraries) are HNF1A wild type cells (WT). In addition, to increase the sequencing depth, each batch was sequenced from 7 flowcells. The raw fastq files were processed via 10X Cell Ranger pipeline (cellranger-2.1.1). To note that GRCh38 reference genome was applied for the whole analysis. "cellranger count" with "–expect-cells = 3000" was run for each library, respectively, and "cellranger aggr" with default parameters was run to aggregate results generated from multiple libraries. A normalized umi-count matrix for all the cells was then generated. As for the quality control, cell and gene filtering process, cells with umi-

count <5000 and umi-count >40,000 and cells with <2000 detected genes were removed. Also, genes with <~1% (25) non-zero counts across all the cells were eliminated from the analysis. To get rid of the Mitochondria cells, cells with more than 10% Mitochondria genes' count were also removed. All the downstream analysis was done with Seurat package in R-3.4.1. Every batch is an independent differentiation (biological replicates). $n = 2$ for each genotype.

**Cell transplantation and in vivo assays**. After 27–30 days of differentiation, 50 μg of stem cell-derived pancreatic cell clusters (~180 islet-like clusters of 300–350 μm diameter; each cluster comprised ~10,000 cells in total) including all islet endocrine cell types were suspended in 30 μl of ice-cold Matrigel (Fisher Scientific, 354277) and loaded into ice-cold 1 ml syringe with a 21 G needle for transplantation. 6–10 weeks old male immunodeficient NSG (NOD.Cg-Prkdcscid Il2rgtm1Wjl/SzJ, Stock No. 005557, The Jackson Laboratories) mice were transplanted in the ventral and medial muscles (medial thigh and posterior lower) of the left thigh. Every 2 weeks post-transplantation, human C-peptide and human insulin were determined in the plasma of recipient mice at a fed state (morning) for 30 weeks. An intraperitoneal glucose tolerance test was performed 18–24 weeks (before STZ injection) and 28–30 weeks (after STZ injection) post-transplantation by injecting 2 g/kg body weight of 20% D-glucose (Sigma, G8270) in PBS overnight fasting (16–18 h). For glibenclamide (Sigma, G0639) administration, 1 mg/kg body weight of the drug diluted in PBS was i.p. injected 24–28 weeks (before STZ injection) post-transplantation, all mice were at fed state but the food was retrieved during the experiment. Blood was collected by tail bleeding in heparinized Eppendorf tubes at time points; plasma was isolated by centrifugation for 15 min at 2000 × g at 4 °C. Human C-peptide was measured using Mercodia Ultrasensitive C-peptide ELISA (10-1141-01) and human insulin using Mercodia Insulin ELISA (10-1113-01) kit according to the manufacturer's protocol. All samples were handled the same way. The insulin detection limit is 1 mU/L as determined by the methodology described in the kit. Blood glucose levels were measured using a glucometer (FreeStyle Lite) and HbA1C using Siemens DCA 2000 Vantage Reagent kit (Siemens, 5035C). To induce diabetes, mice were i.p. injected for 5 consecutive days with 40 mg/kg body weight of Streptozotocin (STZ) (Sigma-Aldrich, S0130-1G) in PBS 24–28 weeks post-transplantation. All experimental procedures on mice were performed according to Columbia University-approved IACUC protocols.

**Bioluminescence and fluorescence imaging**. NSG mice transplanted with $GAPDH^{Luciferase/wt}$ and $INS^{GFP/wt}$ double reporter hESCs lines were i.p. injected with 150 mg/kg body weight of D-luciferin potassium salt (Gold Biotechnology, luck-2G) in PBS 15 min before imaging on a IVIS spectrum optical imaging system (PerkinElmer). Signals were acquired with 1-min exposure and analyzed using the Living image analysis software (Xenogen Corp.). Circular regions of interest (ROI) of the same size for all experiments were drawn around the signal in the left thigh and photons emitted over the time of exposure within the ROI measured. Bioluminescence was measured every 2 weeks post-transplantation for 30 weeks and after isolation of the graft at that time point. Luminescence was measured as described for bioluminescence after isolation of the graft. Background signals were subtracted from a nearby region and were consistent over time.

**Electron microscopy**. Fixed cells/tissues were incubated with 2.5% glutaraldehyde and 2% paraformaldehyde in 100 mM sodium cacodylate buffer (pH 7.4) overnight. Samples were then treated with 1% osmium tetroxide in 100 mM sodium cacodylate buffer for 1 h, washed in distilled water four times (10 min/wash), and then treated with 2% aqueous uranyl acetate overnight at 4 °C in the dark. Samples were washed and sequentially dehydrated with increasing concentrations of acetone (20%, 30%, 50%, 70%, 90%, and 100%) for 30 min each, followed by three additional treatments with 100% acetone for 20 min each. Samples were then infiltrated with increasing concentrations of Spurr's resin (25% for 1 h, 50% for 1 h, 75% for 1 h, 100% for 1 h, 100% overnight at room temperature), and then incubated overnight at 70 °C in a resin mold. Sections of 50–90 nm were cut on a Leica ultramicrotome with a diamond knife. Imaging was performed on an FEI Talos L120C operating at 120 kV.

**Immunogold electron microscopy**. 70 nm-thick sections from the embedded samples were placed onto carbon formvar 75 mesh nickel grids and etched using 4% sodium metaperidodate for 10 min before being washed twice in distilled water and then blocked for 1 h. Grids were incubated with the primary C-peptide antibody (Supplementary Data 7) (1 in 10 dilutions) for 1 h at 4 C overnight. The next day grids underwent seven washes in 1xPBS and then incubated in anti-rat 6 nm gold secondary (1 in 50 dilutions) for 1 h (Supplementary Data 7). After this, the grid was washed seven times in 1xPBS and twice in distilled water. Samples were then imaged on a Thermo Fisher Talos L120C operating at 120 kV.

**Statistics and reproducibility**. All statistics were performed using Prism GraphPad software (La Jolla, CA). Normality of data set distributions was assessed by Shapiro–Wilk test and D'Agostino and Pearson omnibus tests. Normally distributed data were analyzed by unpaired two-tailed t-test. For data not normally distributed we used a two-tailed Mann–Whitney test. Data were expressed as mean ± standard error of the mean (SEM). $p < 0.05$ was considered

statistically significant. *$p < 0.05$, **$p < 0.01$, ***$p < 0.001$. n.s. indicates a non-significant difference.

**Reporting summary**. Further information on research design is available in the Nature Research Reporting Summary linked to this article.

## Data availability

Single-cell RNA sequencing and bulk RNA-sequencing data were deposited in NCBI's Gene Expression Omnibus (GEO) database and the accession number is GSE128331 and GSE129653. Additional data from Haliyur et al. [29] GSE116559 were used. Unprocessed blots can be found in Fig. S14 and source data is available in Supplementary Data 8.

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

## Acknowledgements

We would like to thank Giacomo Diedenhofen for help with western blots and Qian Du for help with differentiation and flow cytometry. This research was supported by the American Diabetes Association (grant #1-16-ICTS-029) and the NYSTEM IDEA award # C029552, Leona and Harry Helmsley Charitable Trust, Helmsley Trust Diabetes Cell Repository, the NYSCF-Robertson award from the New York Stem Cell Foundation, the Sanofi iAwards, and the Naomi Berrie Foundation program for Cellular Therapies of Diabetes. These studies used the resources of the Herbert Irving Comprehensive Cancer Center Flow Cytometry Shared Resources funded in part through Center Grant P30CA013696 and the Diabetes and Endocrinology Research Center Flow Core Facility funded in part through DRC Center Grant (5P30DK063608) and the MBMG Core in the New York Nutrition and Obesity Research Center (5P30DK026687).

## Author contributions

B.J.G. designed the studies, performed differentiation, transplantation of all cell lines and downstream experiments, gathered the data, analyzed the data, and wrote the paper with input from authors. D.E. designed the studies, discussed results, reviewed and edited the manuscript. D.E. performed skin biopsy stem cell derivation. J.G. performed single-cell RNA sequencing and B.J.G., H.Z., J.L., and Y.S. designed, analyzed, and compiled the single-cell RNA-sequencing data. B.J.G. and C.N.G. performed EMC imaging. W.K.C. provided genetic sequence information of HNF1A-MODY subjects. Y.X., Y.W., and J.O. performed microfluidic perfusion and generated data. M.H.B. performed flow cytometry of HNF1A-WT and HNF1A-KO. J.N., D.J.W., and H.M.C. performed calcium imaging, measurements, and data interpretation. X.C. contributed to human islets. R.S.G. performed oversight of human subject research and provided de-identified clinical information. D.E., C.A.L., and R.L. provided guidance in the design and interpretation of studies.

## Competing interests

The authors declare no competing interests.
