## [Peer Review File · Communications Biology]

Reviewers' comments:

Reviewer #1 (Remarks to the Author):

This manuscript describes using the in vitro differentiation of human pluripotent stem cells to model MODY3 caused by mutations in HNF1A. The authors generate several gene edited lines in ESCs or from a MODY3 patient's iPSC line that was gene corrected. They go on to examine endocrine cell development and function in vitro and as well as after transplantation in a mouse model. They find defects in gene expression, a skewing towards alpha cells and defects in calcium signaling and insulin secretion. Finally, they genome edit several HNF1A targets, including PAX4, CACNA1A and SYT13 to determine how each contributes to the HNF1A phenotype. Overall, this manuscript confirms previously published findings about the role of HNF1A in endocrine cell development and extends them to look more closely at a set of HNF1A targets and perform more detailed in vivo studies. See below for specific concerns.

Major Concerns:

1. There are some major issues with flow cytometry throughout the manuscript. In 2D the graphs look to be improperly compensated. Flow cytometry in 3 and S3L (WT) are not convincing with barely any detectable c-peptide signal, much worse than the staining seen in figure 2. In addition, there are practically no cells on the plot in S3. In S6C, the HNF1A KO1 appears to have practically all of the cells GCG single positive while for KO2 looks more similar to the WT with slightly more GCG expression but mainly in the C-peptide+ population. In 6D the percentages stated in the graph don't seem to match with what the plots actually look like with virtually no events in the KO3 sample. Similar issues in G. 6I in the KO shows 20% c-peptide+ cells, looks like essentially none to me. 6J is the worst, there are plots with percentages but no actual cells visible on the plots. 7SP also has issues with the HNF1A KO FACS plot. S10A also has very poor uninterpretable data. As so much of the quantification is using flow cytometry to calculate percentages of cells expressing various markers it makes me worry about the validity of these numbers. These studies should be examined by an expert in flow cytometry to assure proper gating and quantification has been performed. Confirming some of these changes by QRT-PCR could also help support these findings.

2. The authors state that the that the HNF1A KO beta cells do not see a decrease in insulin content. This is somewhat surprising considering that in both the mouse KO and the other publication looking at the human stem cell model saw a decrease in the homozygous KO. I was also very surprised that the authors had similar insulin content in primary human islets compared to ESC derived beta cells. Generally insulin content in ESC derived beta cells is significantly lower than what is seen in primary islets. This makes me a little wary of these data and how this number was calculated. This data should be supported by other assays looking at either protein and/or RNA. For example QRT-PCR for insulin on sorted beta cells, MFI of insulin or c-peptide by flow cytometry, gated on INS+ cells, or western blot etc...

3. Following point 2, while the insulin content findings were different, the majority of the findings from this work including endocrine cell development including alpha cell skewing, and many gene targets are similar to 2 prior publications in this area. First was a work by the Powers lab examining endocrine cells from a single individual with MODY3 and second was work from the Gadue lab using a similar stem cell model as this manuscript. Another difference that the authors bring up is the lincRNA target found by the Gadue lab. These authors do not see a change in their stem cell lines and they state that examination of the Powers dataset show the lincRNA was not dysregulated in their MODY3 patient's beta cells. Looking back at these papers seems to indicate the differences are not as large as suggested. For example, the Gadue group only found the lincRNA differences in one of the 2 stem cell lines they examined. In addition, they also published a graph from the Powers' dataset looking at the LincRNA which is essentially the same as Figure S5M. They also included data from MODY3 alpha cells and did see a downregulation of the lincRNA here. The last difference seen between the papers was basal insulin secretion. Both the Powers

and Gadue papers saw increased or similar basal secretion with failure of increased secretion upon stimulation with high glucose. The authors should discuss the differences between their work and the published literature with potential thoughts on what they might imply about either the model or the particular biology they are studying. Addressing this I think is important for the field of stem cell disease modeling. Please include insulin content in this discussion as well. Discussing whether these differences are due to how the experiments were performed, differential genetic backgrounds or true differences in biology would be important when deciding about the relevance of the stem cell model system and its ability to recapitulate beta cell disease.

4. In figure 4 the authors talk about altered insulin to c-peptide stoichiometry since human insulin was barely detectable in the blood of transplanted mice in the KO while the c-peptide was easily detectable but at significantly lower levels compared to the WT. Is this an important finding that reveals relevant biology? Could this not be due to the fact that the KO are secreting much less insulin/c-peptide in general and that insulin will be consumed by the animal while c-peptide will not and can accumulate more easily? Would not the fact that treatment with Glib which rescues the secretion defect also rescues the ability to detect insulin in the plasma support this interpretation? Its not clear to me what the authors are trying to show with these data. Are they trying to say that insulin and c-peptide are secreted differentially in the KO?? If so the data are not convincing. A perfusion system would be better able to detect differences in secretion of c-peptide vs insulin without the confounding effects of insulin consumption by resident cells/tissue. Alternatively, this data could be excluded from the manuscript or the alternative (and more likely) interpretation of insulin consumption could be proposed as a secondary explanation for their data.

Reviewer #2 (Remarks to the Author):

This manuscript represents an enormous amount of work where the authors have used multiple cellular models to understand the effect of HNF1A haploinsufficiency or point mutations on pancreatic islet cell development and function. They have approached this using CRISPR-cas genome editing in human ES cells coupled with in vitro differentiation and maturation of the SC-beta cells in a murine system. They have also compared their findings with patient derived IPS cells from individuals with pathogenic HNF1A mutations causing HNF1A-MODY diabetes. The authors have gone beyond transcriptomic studies (bulk and single cell RNA-seq) to functionally characterize the effects of the mutations on insulin secretion and calcium levels within the cells. The work is largely well performed, with additional lines (e.g. PAX4 KO) generated to solidly and support hypothesizes arising from the transcriptomic data. There are however a number of areas where the authors could improve their presentation of their data and below I list areas where I have some concerns which I would recommend they address.

The strengths of the study are:

- * Use of both isogenic ES lines and patient derived IPS cel lines
- * Maturation of the SC derived beta-cells in a mouse
- * Functional as well as transcriptomic data
- * The manuscript adds important and valuable data to the growing body of work studying the effects of transcription factor mutations on islet-cell development using human stem cell models.

The weaknesses of the study are:

- * The limits of the functional follow up on effects on insulin exocytosis
- * The fact several other manuscripts reporting the effects of HNF1A mutations/loss in stem cell models have been reported.
- * Lack of human data from carriers of HNF1A variants to support the observations that there is an altered ratio of Insulin/c-peptide.

Comments for the Authors

- * Please do not use the term MODY3 - this is an obsolete term and should be replaced by HNF1A-

MODY. The term MODY3 was used PRIOR to the identification of the genetic cause.

* Abstract: The statement that the studies provide promise for cell based therapies for HNF1A-MODY seems an unnecessary stretch.

* The references cited for the number of genetic causes of MODY and the proportion of MODY cases due to HNF1A mutations are outdated. A recent review by Toni Pollin who heads the Clin Gen Monogenic Diabetes Curation Panel would be a good place to find the most recent and importantly widely accepted information (Zhang et al JCI 2021).

* Please update your article to include the recent paper in Nature Communications from Adrian Teo's group in Singapore. Please discuss how your findings support and extend their observations.

* The section on HNF1A-deficiency affecting glucose mediated insulin granule release needs revisiting. The authors have not directly measured or assessed insulin granule release. To do this they would need to perform more sophisticated in vitro assays to measure exocytosis (capacitance measurements). In the absence of these data the authors should temper their speculation regarding the mechanism for the observed differences in insulin secretion at low and high glucose levels. Their hypothesis is reasonable but they have not actually tested it or demonstrated an effect.

* The observations relating to the ratio of insulin to c-peptide are interesting. It would be extremely helpful to understand whether these observations mirror what happens in vivo or if they are an artifact of the in vitro study. In the absence of data from patients with HNF1A MODY mutations can the authors use publicly available data from perhaps the MAGIC consortium to explore whether common variation at this locus is associated with insulin to c-peptide ratio in the general population?

* The observations regarding insulin granule structure/density are interesting. It is not clear to this reviewer though how the authors can state that the abnormal granules are not secreted in response to glucose and only to sulphonylureas?

* I have some concerns regarding the conclusions being drawn from the IPS cell studies. Although I am very open to different HNF1A-MODY mutations having different severities which could manifest with varying degrees on cellular phenotypes I feel that the lack of a comparator in the IPS cell line (a mutation causing HNF1A haploinsufficiency which would mirror the ES cell model) prevents the conclusion being drawn. It is not unexpected that the point mutation would result in a mutated protein rather than haploinsufficiency. However, the authors have not compared like for like. They need to study the effect of haploinsufficiency in the IPS cell model or the point mutation in the ES cell model to ensure that their conclusions are not related to differences in the experimental model.

* Several of the figures are very busy and contain elements which are too small to see

* The manuscript contains multiple themes and narratives and at times is challenging for the reader to follow.

Reviewer #3 (Remarks to the Author):

This study investigates the effect of mutated HNF1A gene on a specific form of diabetes, Maturity Onset Diabetes of the Young (MODY3). The molecular mechanisms caused by HNF1A deficiencies were examined in vitro with CRIPSR knockout studies of hESCs, following in vitro differentiation towards pancreatic endocrine subsets. The authors convincingly analyzed and concluded that HNF1A deficiency is linked to altered calcium homeostasis in the cell, disturbed exocytosis of

insulin secretory vesicles and thus altered ratio of insulin/C-peptide release.

Beside molecular and functional characterization of the in vitro generated endocrine cells, they also present several profound in vivo mouse experiments. On top, they linked the results to clinical relevance by establishing a disease model with iPSC of patients harboring HNF1A mutations.

The authors persuasively outlined their results and conclusion by a huge variety of experiments, including in vitro and in vivo, as well as patient-specific studies. In this work the profound stem cell-based disease models and the molecular mechanisms induced by HNF1A deficiency can contribute to a better understanding of the MODY3 disease.

Since the here presented results are very convincing, only some comments of typos and illustration for a minor revision as follows:

1. Check that all subfigure numberings (and subfigures in general) are better aligned (e.g. Figure 2), otherwise it is getting slightly confusing due to lots of images in one figure
2. Figure 4B+C, Figure 6F, Figure S8H: legend is missing
3. Figure 5A: add scale bars (missing for some images)
4. Figure 5A: Typo \diamond microscopy (also Suppl. Figure 9)
5. Check the bar graphs again for adding single measurements (dots) \diamond done for most but not all bar graphs
6. Uniformly decide if you spell out numbers, e.g. "30 weeks" or "thirty weeks"
7. Try to avoid beginning sentences with numbers, sometimes it is then more difficult to read
8. Material/methods section: uniformly use the same tense (past)
9. Figure Suppl. 3 L.: seem to be very few events measured for flow cytometry (only for wildtype), replace, if possible \diamond please show the same number of events in all plots to make them more comparable
- Figure Suppl. 6J: almost no events at all can be seen for the Flow Cytometry plots, thus inconclusive and quantification does not make any sense
10. Consider including single channel pictures for your fluorescent images (e.g. Figure 1 A)
11. Fluorescent channels are often saturated. If available, please use pictures taken with lower laser / exposure.

Reviewers' comments:

Reviewer #1 (Remarks to the Author):

This manuscript describes using the *in vitro* differentiation of human pluripotent stem cells to model MODY3 caused by mutations in HNF1A. The authors generate several gene edited lines in ESCs or from a MODY3 patient's iPSC line that was gene corrected. They go on to examine endocrine cell development and function *in vitro* and as well as after transplantation in a mouse model. They find defects in gene expression, a skewing towards alpha cells and defects in calcium signaling and insulin secretion. Finally, they genome edit several HNF1A targets, including PAX4 CACNA1A and SYT13 to determine how each contributes to the HNF1A phenotype. Overall, this manuscript confirms previously published findings about the role of HNF1A in endocrine cell development and extends them to look more closely at a set of HNF1A targets and perform more detailed *in vivo* studies. See below for specific concerns.

Major Concerns:

1. There are some major issues with flow cytometry throughout the manuscript. We have addressed the flow plot issues in the main Figures 2D and 3F by adding new flow plots in Fig. 2D, and adjusting the gating in Fig. 3F. Some of the flow plot data in supplementary figures where cell counts were considered too low, and where redundant data from other analysis (immunohistochemistry, scRNA seq were available), were removed, and the text was altered accordingly. As newly constituted, the flow cytometry data are treated as supportive of data presented in histology (Fig. 2A), and single cell sequencing (Fig. 2E).

In 2D the graphs look to be improperly compensated.

Response: We have replaced the flow cytometry plots in **Figure 2D** (see also Figure S3M) with new plots from differentiated cells. Importantly, we now include negative controls that use both primary and secondary antibody staining for gating of negative cells (Fig. S3M). Using this new differentiation and gating strategy, we find a doubling of glucagon-positive endocrine cells in *HNF1A* knockout genotypes. Please also note that this conclusion of increased glucagon-positive cells is based on 3 different and independent assays, immunostaining and microscopy conducted on differentiated pancreatic cells *in vitro* and *in vivo* in grafts, and single cell RNA seq of HNF1A KO cells. These findings are fully consistent with recent observations of increased proportions of alpha cells in the islets of a patient with MODY 3, segregating for a null mutation in *HNF1A* (Cardenas-Diaz et al., 2019). Other aspects of the important congruence of our results with those reported by Cardenas-Diaz are discussed below.

Flow cytometry in 3 and S3L (WT) are not convincing with barely any detectable c-peptide signal, much worse than the staining seen in figure 2. In addition, there are practically no cells on the plot in S3.

We have improved the gating in **Fig. 3F** in reference to the negative control. Adjusting of the gates did not change our conclusion that that differentiation efficiency is not affected in CACNA1 or SYT13 mutants. Furthermore, in **Figure 3E** insulin-GFP fluorescence is indistinguishable from controls, consistent with our conclusion that these mutations does not affect beta cell differentiation per se.

Figure S3L has been removed because **Fig. 2A-E**, as well as **Fig. S3I-S3M** already demonstrate the developmental bias using 3 different assays and different *HNF1A* KO clones.

In S6C, the HNF1A KO1 appears to have practically all of the cells GCG single positive while for KO2 looks more similar to the WT with slightly more GCG expression but mainly in the C-peptide+ population. In 6D the percentages stated in the graph don't seem to match with what the plots actually look like with virtually no events in the KO3 sample. Similar issues in G. 6I in the KO shows 20% c-peptide+ cells, looks like essentially none to me. 6J is the worst, there are plots with percentages but no actual cells visible on the plots.

These comments apparently refer to **Fig. S6** (not Figure 6). Flow plots of **Fig. S6** have been removed because of low cell numbers, and the text amended accordingly. The text regarding cell proliferation and Tunel staining has also been eliminated. These points are not central to the narrative of the manuscript. The emphasis on the relevant biology calcium, insulin granules, and insulin secretion as well as *in vivo* function help simplify the narrative.

7SP also has issues with the HNF1A KO FACS plot.

Figure S7O and **Fig. S7P** were removed and the data and text regarding the *Pax4* mutation were also removed from the manuscript. The *Pax4* mutation confers a developmental bias towards alpha cells that has been well documented in studies in mice.

S10A also has very poor uninterpretable data. As so much of the quantification is using flow cytometry to calculate percentages of cells expressing various markers it makes me worry about the validity of these numbers. These studies should be examined by an expert in flow cytometry to assure proper gating and quantification has been performed. Confirming some of these changes by QRT-PCR could also help support these findings.

We show that STZ has no differential effect on insulin secretion in HNF1A knockout cells compared to controls (now **Fig. 10A**). The flow cytometry data of **Fig. S10A** have been removed, and the text amended accordingly.

2. The authors state that the that the HNF1A KO beta cells do not see a decrease in insulin content. This is somewhat surprising considering that in both the mouse KO and the other publication looking at the human stem cell model saw a decrease in the homozygous KO. I was also very surprised that the authors had similar insulin content in primary human islets compared to ESC derived beta cells. Generally insulin content

in ESC derived beta cells is significantly lower than what is seen in primary islets. This makes me a little wary of these data and how this number was calculated. This data should be supported by other assays looking at either protein and/or RNA. For example QRT-PCR for insulin on sorted beta cells, MFI of insulin or c-peptide by flow cytometry, gated on INS+ cells, or western blot etc...

Response: We believe the reviewer is referring primarily to supplemental **Figure S8B**. To address the reviewer's concerns regarding our analysis of primary islets, we removed the primary islet data point in **Fig. S8B**. This ensures that only samples that can be physically dissociated in an equivalent manner are being compared. The main point of this Figure is the comparison of WT and *HNF1A* knockout cells, which shows no difference in insulin content. Insulin content of primary human islets was not mentioned in the text, and no changes to text needed to be made. The reviewer is correct that insulin content is lower in stem cell derived beta-like cells. We have reported such comparison in a previous study for the types of cells used here (Sui et al., 2017).

3. Following point 2, while the insulin content findings were different, the majority of the findings from this work including endocrine cell development including alpha cell skewing, and many gene targets are similar to 2 prior publications in this area. First was a work by the Powers lab examining endocrine cells from a single individual with MODY3 and second was work from the Gadue lab using a similar stem cell model as this manuscript. Another difference that the authors bring up is the lincRNA target found by the Gadue lab. These authors do not see a change in their stem cell lines and they state that examination of the Powers dataset show the lincRNA was not dysregulated in their MODY3 patient's beta cells. Looking back at these papers seems to indicate the differences are not as large as suggested. For example, the Gadue group only found the lincRNA differences in one of the 2 stem cell lines they examined. In addition, they also published a graph from the Powers' dataset looking at the LincRNA which is essentially the same as Figure S5M. They also included data from MODY3 alpha cells and did see a downregulation of the lincRNA here. The last difference seen between the papers was basal insulin secretion. Both the Powers and Gadue papers saw increased or similar basal secretion with failure of increased secretion upon stimulation with high glucose. The authors should discuss the differences between their work and the published literature with potential thoughts on what they might imply about either the model or the particular biology they are studying. Addressing this I think is important for the field of stem cell disease modeling. Please include insulin content in this discussion as well. Discussing whether these differences are due to how the experiments were performed, differential genetic backgrounds or true differences in biology would be important when deciding about the relevance of the stem cell model system and its ability to recapitulate beta cell disease.

The reviewer raises important points regarding differences in results among these 2 prior publications and our's. While we do not find a downregulation of lincRNA (based on Cardenas-Diaz' own data as well as ours), we do concur with regard to a developmental bias to alpha cells due to *HNF1A* deficiency. Cardenas-Diaz paper's conclusions is based on differences in the expression of that lincRNA LINC01139 and

its effect on mitochondrial respiration in haploinsufficient cells. In contrast to this study, the study by Gadue and colleagues was also conducted *in vitro*, on cells that may not have fully mature mitochondrial function. Surprisingly, virtually all phenotypes were as strong or stronger in heterozygous *HNF1A* null cells than in homozygous knockout cells. This series of observations is not reproduced in our study, and difficult to explain, because Cardenas-Diaz introduced the mutation at the start codon, avoiding the production of a mutant protein that could interfere with the wild type HNF1A protein or perhaps HNF1B. Thus, we believe strongly that it is important to provide a complementary dataset with emphasis on calcium phenotypes and insulin secretion and *in vivo* functionality of HNF1A deficient cells, which is more consistent with the treatment of HNF1A-MODY.

We now dedicate a paragraph in the Discussion to these prior publications:

“Our study illustrates both congruence as well as differences with other stem cell models of HNF1A deficiency. Our data and another study show a bias in endocrine differentiation to glucagon positive cells in HNF1A knockout cells (46). Cardenas-Diaz also document this bias in heterozygous cells, to an extent comparable to biallelic knockout cells. Mutations in *HNF1A* introduced by Cardenas-Diaz are located adjacent to the start codon, excluding possible dominant-negative effects of a truncated protein that could interfere with the wild type protein. Such dominant negative truncating mutations were made by another group and shown to affect pancreatic differentiation from pluripotent stem cells by interfering with *HNF1B* function (47). Both our study and Cardenas-Diaz identify a deficiency of glucose--stimulated insulin secretion in *HNF1A* biallelic knockout cells (46). Surprisingly, Cardenas-Diaz and colleagues found that heterozygous and homozygous knockout cells were equally impaired for *in vitro* insulin secretion. The finding that haploinsufficiency was as detrimental as a complete knockout or a dominant negative mutation is surprising. Cardenas-Diaz linked insulin secretion phenotypes to altered mitochondrial function, decreased expression of *LINC01139* and increased expression of LDHA involved in anaerobic glycolysis, again reporting similar reduction in expression of these genes in both heterozygous and homozygous mutant cells. An increase of LDHA due to HNF1A deficiency was not observed in our study, neither in bulk-RNA seq nor in single cell RNA seq, and neither in KO cells or heterozygous cells. Neither did Haliyur see this effect in HNF1A^{T260M} heterozygous pancreatic islets (30). LDHA is highly expressed in undifferentiated pluripotent stem cells; its expression may thus be affected by the maturity of stem cell-derived pancreatic cell clusters. An impairment of glucose stimulated insulin-secretion after transplantation *in vivo* was also reported by (48) in HNF1A^{H126D} heterozygous patient cells, though differences in comparison wild type cells were variable, and not statistically significant (48). We used patient cells heterozygous for nonsense HNF1A^{R220Q} located in the DNA binding domain. In these HNF1A^{R220Q} heterozygous cells, we found no defect in stimulated insulin secretion *in vitro* and after transplantation *in vivo*. Only after months of *in vivo* evaluation did these cells start to show reduced insulin secretion.

An intriguing concordance of phenotypes of a stem cell model (Cardenas-Diaz) and islets from a HNF1A-MODY patient (Haliyur) is the increased basal secretion rate of insulin. Interestingly, we find that *HNF1A* knockout cells show reduced expression of

KCNH6, mutations in which are associated with elevated insulin secretion (49). However, phenotypic concordances among studies are not necessarily due to concordant mechanisms. Other factors, including increased *LDHA* expression and rates of glycolysis may also affect basal insulin secretion in stem cell-derived insulin producing cells. In a further difference to Cardenas-Diaz et al., we did not detect changes in *LINC01139* expression in stem cell derived beta-like cells. *LINC01139* was reported to affect mitochondrial function in beta cells. Another study has found downregulation of glucose transporters and reduced ATP generation in patient iPS derived beta cells in comparison to control wild type lines (48), while we found downregulation of genes involved in calcium homeostasis.

In summary, this as well as other stem cell-based studies on *HNF1A* deficiency demonstrate the utility of stem cell-based models in defining the molecular physiology of β -cell failure in gene-specific diabetes in humans. Specifically, they reveal pleiotropic effects of *HNF1A* deficiency at several levels of β -cell biology and function, consistent with the diversity of its transcriptional targets.

4. In figure 4 the authors talk about altered insulin to c-peptide stoichiometry since human insulin was barely detectable in the blood of transplanted mice in the KO while the c-peptide was easily detectable but at significantly lower levels compared to the WT. Is this an important finding that reveals relevant biology? Could this not be due to the fact that the KO are secreting much **less insulin/c-peptide in general and that insulin will be consumed by the animal while c-peptide will not and can accumulate more easily?**

Response: This is a very important point. Our key argument against this scenario is that c-peptide levels are sufficiently high in the KO to anticipate detection of insulin if the two peptides were secreted concordantly (See **Figure 4A below**). At week 30, based on c-peptide levels, there should be as high insulin in the knockout as at week 18 in the wild type cells, about 10mU/L. We see none. This strongly points to a bias in secretion. Furthermore, we also detect a bias in the stoichiometry of insulin and c-peptide release in vitro (Fig. S8I).

Nevertheless we now include a statement in the discussion section to indicate that other possibilities remain: "We cannot formally exclude the possibility that insulin secreted from mutant cells is more efficiently removed from the circulation than insulin secreted from wild type cells."

From Figure 4. ***HNF1A* deficiency alters the stoichiometry of insulin to C-peptide secretion *in vivo*.** [why don't you express insulin in moles to make comparison relative concentrations more explicit.

Would not the fact that treatment with Glib which rescues the secretion defect also rescues the ability to detect insulin in the plasma support this interpretation? Its not clear to me what the authors are trying to show with these data. Are they trying to say that insulin and c-peptide are secreted differentially in the KO?? If so the data are not convincing. A perfusion system would be better able to detect differences in secretion of c-peptide vs insulin without the confounding effects of insulin consumption by resident cells/tissue. Alternatively, this data could be excluded from the manuscript or the alternative (and more likely) interpretation of insulin consumption could be proposed as a secondary explanation for their data.

Response: We removed the conclusion of bias from the title. We agree with the reviewer that other possible interpretations should be discussed, which we do as indicated above in the Discussion. We also removed the altered stoichiometric secretion from the one-sentence summary and the abstract, and limit relevant text to the Discussion.

We do, however, wish to include the data in the manuscript, as they are unique in the field, and the availability of these data will likely lead others to efforts to confirm or refute the inference.

Reviewer #2 (Remarks to the Author):

This manuscript represents an enormous amount of work where the authors have used multiple cellular models to understand the effect of HNF1A haploinsufficiency or point mutations on pancreatic islet cell development and function. They have approached this using CRISPR-cas genome editing in human ES cells coupled with in vitro differentiation and maturation of the SC-beta cells in a murine system. They have also compared their findings with patient derived IPS cells from individuals with pathogenic HNF1A mutations causing HNF1A-MODY diabetes. The authors have gone beyond transcriptomic studies (bulk and single cell RNA-seq) to functionally characterize the effects of the mutations on insulin secretion and calcium levels within the cells. The work is largely well performed, with additional lines (e.g. PAX4 KO) generated to solidly and support hypothesizes arising from the transcriptomic data. There are however a number of areas where the authors could improve their presentation of their data and below I list areas where I have some concerns which I would recommend they address.

The strengths of the study are:

- * Use of both isogenic ES lines and patient derived IPS cell lines
- * Maturation of the SC derived beta-cells in a mouse
- * Functional as well as transcriptomic data
- * The manuscript adds important and valuable data to the growing body of work studying the effects of transcription factor mutations on islet-cell development using human stem cell models.

The weaknesses of the study are:

- * The limits of the functional follow up on effects on insulin exocytosis
- * The fact several other manuscripts reporting the effects of HNF1A mutations/loss in stem cell models have been reported.
- * Lack of human data from carriers of HNF1A variants to support the observations that there is an altered ratio of Insulin/c-peptide.

Response: We agree with your comments and indications of weaknesses. We have endeavored to rectify these issues below.

Comments for the Authors

- * Please do not use the term MODY3 - this is an obsolete term and should be replaced by HNF1A-MODY. The term MODY3 was used PRIOR to the identification of the genetic cause.

Response: Done

* Abstract: The statement that the studies provide promise for cell based therapies for HNF1A-MODY seems an unnecessary stretch.

Response: We now more specifically state for whom such treatments could be useful in the discussion section: “While sulfonylureas can be used to treat patients with HNF1A-MODY effectively, insulin dependence is common after years of treatment. For these patients, cell therapy might be considered .”

* The references cited for the number of genetic causes of MODY and the proportion of MODY cases due to HNF1A mutations are outdated. A recent review by Toni Pollin who heads the Clin Gen Monogenic Diabetes Curation Panel would be a good place to find the most recent and importantly widely accepted information (Zhang et al JCI 2021).

Response: we now cite this reference and adapt the numbers accordingly in the introduction.

* Please update your article to include the recent paper in Nature Communications from Adrian Teo's group in Singapore. Please discuss how your findings support and extend their observations.

Response: We have included this paper in the Discussion (see above in response to Reviewer 1). The studies by Teo's group highlight glucose transporter GLUT2 as an *HNF1A* target gene, which is likely responsible for aspects of the disease. We did not detect a significant difference in glucose transporter GLUT2 expression in stem cell derived beta-like cells. Teo's group used non-isogenic cell lines, while we used isogenic lines. Because the studies focus on different downstream target genes, the two datasets are highly complementary.

* The section on HNF1A-deficiency affecting glucose mediated insulin granule release needs revisiting. The authors have not directly measured or assessed insulin granule release. To do this they would need to perform more sophisticated in vitro assays to measure exocytosis (capacitance measurements). In the absence of these data the authors should temper their speculation regarding the mechanism for the observed differences in insulin secretion at low and high glucose levels. Their hypothesis is reasonable but they have not actually tested it or demonstrated an effect.

Response: The reviewer is correct. We have tempered our inferences regarding this point and expanded our discussion of studies needed to definitively assess it (see above in response to Reviewer 1).

* The observations relating to the ratio of insulin to c-peptide are interesting. It would be extremely helpful to understand whether these observations mirror what happens in vivo or if they are an artifact of the in vitro study. In the absence of data from patients with HNF1A MODY mutations can the authors use publicly available data from perhaps the MAGIC consortium to explore whether common variation at this locus is associated with

insulin to c-peptide ratio in the general population?

Response: We do have clinical data, which we plan to expand into a more complete study. These are consistent with our stem cell data, but were collected in an uncontrolled setting and are not currently suitable for publication. Because of this, we limit the discussion of the insulin/c-peptide ratio to the discussion section and mention possible alternative interpretations.

* The observations regarding insulin granule structure/density are interesting. It is not clear to this reviewer though how the authors can state that the abnormal granules are not secreted in response to glucose and only to sulphonylureas?

Response: Thank you for making this point. We now revised our statements to what we measure: the secretion of insulin (as opposed to the exocytosis of granules).

* I have some concerns regarding the conclusions being drawn from the IPS cell studies. Although I am very open to different HNF1A-MODY mutations having different severities which could manifest with varying degrees on cellular phenotypes I feel that the lack of a comparator in the IPS cell line (a mutation causing HNF1A haploinsufficiency which would mirror the ES cell model) prevents the conclusion being drawn. It is not unexpected that the point mutation would result in a mutated protein rather than haploinsufficiency. However, the authors have not compared like for like. They need to study the effect of haploinsufficiency in the IPS cell model or the point mutation in the ES cell model to ensure that their conclusions are not related to differences in the experimental model.

Response: These are valid concerns. In the heterozygous iPSC model, we compare isogenic cells with a Crispr-mediated correction of R200Q. These represent an authentic allelic series for an actual human mutation. When transplanted into mice, the mutant cells show gradual deterioration of in-vivo insulin secretion in comparison to the corrected cells. This progression recapitulates the apparent natural history in many HNF1A-MODY patients. That evolved phenotype in the mice resembles that of the hESC null islets described in the manuscript. While a comparable study using an hESC allelic series for a null mutation would be interesting, we think that such an experiment is not critical to the interpretation of the iPSC experiment with cells from an actual patient.

* Several of the figures are very busy and contain elements which are too small to see

Response: We have enlarged the Figures and Figure elements throughout the manuscript.

* The manuscript contains multiple themes and narratives and at times is challenging for the reader to follow.

Response: We have shortened the paper and specifically reorganized the Discussion of the insulin/c-peptide ratio. We have de-emphasized the differentiation bias of mutant *HNF1A* alleles towards glucagon-producing cells, as now published in a recent study. We also consolidated the Discussion of relevant related studies (Cardenas-Diaz et al., Low et al.) to a single paragraph in the discussion section.

Reviewer #3 (Remarks to the Author):

This study investigates the effect of mutated HNF1A gene on a specific form of diabetes, Maturity Onset Diabetes of the Young (MODY3). The molecular mechanisms caused by HNF1A deficiencies were examined *in vitro* with CRISPR knockout studies of hESCs, following *in vitro* differentiation towards pancreatic endocrine subsets. The authors convincingly analyzed and concluded that HNF1A deficiency is linked to altered calcium homeostasis in the cell, disturbed exocytosis of insulin secretory vesicles and thus altered ratio of insulin/C-peptide release.

Beside molecular and functional characterization of the *in vitro* generated endocrine cells, they also present several profound *in vivo* mouse experiments. On top, they linked the results to clinical relevance by establishing a disease model with iPSC of patients harboring HNF1A mutations.

The authors persuasively outlined their results and conclusion by a huge variety of experiments, including *in vitro* and *in vivo*, as well as patient-specific studies. In this work the profound stem cell-based disease models and the molecular mechanisms induced by HNF1A deficiency can contribute to a better understanding of the MODY3 disease.

Response: We thank the reviewer for the very encouraging comments.

Since the here presented results are very convincing, only some comments of typos and illustration for a minor revision as follows:

1. Check that all subfigure numberings (and subfigures in general) are better aligned (e.g. Figure 2), otherwise it is getting slightly confusing due to lots of images in one figure

Response: We have followed this advice throughout the revision. And have de-cluttered the manuscript by removal of some of the figures.

2. Figure 4B+C, Figure 6F, Figure S8H: legend is missing
The legend has been restored.

3. Figure 5A: add scale bars (missing for some images)
All amended as suggested.

4. Figure 5A: Typo \diamond microscopy (also Suppl. Figure 9)
Corrected.

5. Check the bar graphs again for adding single measurements (dots) \diamond done for most but not all bar graphs
Some bars are not displayed with dots because of the large number of biological replicates. In these instances, we have now added the relevant numbers in the Figure legends.

6. Uniformly decide if you spell out numbers, e.g. “30 weeks” or “thirty weeks”

Thank you. We have changed this to “30 weeks” through the manuscript

7. Try to avoid beginning sentences with numbers, sometimes it is then more difficult to read

Amended.

8. Material/methods section: uniformly use the same tense (past)

Amended

9. Figure Suppl. 3 L.: seem to be very few events measured for flow cytometry (only for wildtype), replace, if possible please show the same number of events in all plots to make them more comparable

Response: As noted in our several responses to Reviewer #1, these flow plots and their associated text have been removed and/or replaced. Our key conclusions regarding the effects of HNF1A mutations on beta cell differentiation bias are made through immunostaining and imaging, as well as single cell RNA sequencing. Flow plots in the main Figure 2 and integral to the main conclusion, were replaced a new flow plot in Fig. 2D.

Figure Suppl. 6J: almost no events at all can be seen for the Flow Cytometry plots, thus inconclusive and quantification does not make any sense

Response: As noted above, these flow plots and the associated text have been removed.

10. Consider including single channel pictures for your fluorescent images (e.g. Figure 1 A)

Response: A new figure with single channel picture has been added in **Figure S3L**.

11. Fluorescent channels are often saturated. If available, please use pictures taken with lower laser / exposure.

Response: To address these concerns regarding fluorescent imaging, we have included single channel images in supplemental **Figure S3L** and [done what to improve those alluded to in the suggestion].

References cited above:

- Cardenas-Diaz, F. L., Osorio-Quintero, C., Diaz-Miranda, M. A., Kishore, S., Leavens, K., Jobaliya, C., Stanescu, D., Ortiz-Gonzalez, X., Yoon, C., Chen, C. S., Haliyur, R., Brissova, M., Powers, A. C., French, D. L. & Gadue, P. 2019. Modeling Monogenic Diabetes using Human ESCs Reveals Developmental and Metabolic Deficiencies Caused by Mutations in HNF1A. *Cell Stem Cell*, 25, 273-289.e5.
- Cujba, A. M., Alvarez-Fallas, M. E., Pedraza-Arevalo, S., Laddach, A., Shepherd, M. H., Hattersley, A. T., Watt, F. M. & Sancho, R. 2022. An HNF1 α truncation associated with maturity-onset diabetes of the young impairs pancreatic progenitor differentiation by antagonizing HNF1 β function. *Cell Rep*, 38, 110425.

- Haliyur, R., Tong, X., Sanyoura, M., Shrestha, S., Lindner, J., Saunders, D. C., Aramandla, R., Poffenberger, G., Redick, S. D., Bottino, R., Prasad, N., Levy, S. E., Blind, R. D., Harlan, D. M., Philipson, L. H., Stein, R. W., Brissova, M. & Powers, A. C. 2019. Human islets expressing HNF1A variant have defective beta cell transcriptional regulatory networks. *J Clin Invest*, 129, 246-251.
- Low, B. S. J., Lim, C. S., Ding, S. S. L., Tan, Y. S., Ng, N. H. J., Krishnan, V. G., Ang, S. F., Neo, C. W. Y., Verma, C. S., Hoon, S., Lim, S. C., Tai, E. S. & Teo, A. K. K. 2021. Decreased GLUT2 and glucose uptake contribute to insulin secretion defects in MODY3/HNF1A hiPSC-derived mutant β cells. *Nat Commun*, 12, 3133.
- Sui, L., Danzl, N., Campbell, S. R., Viola, R., Williams, D., Xing, Y., Wang, Y., Phillips, N., Poffenberger, G., Johannesson, B., Oberholzer, J., Powers, A. C., Leibel, R. L., Chen, X., Sykes, M. & Egli, D. 2017. Beta Cell Replacement in Mice Using Human Type 1 Diabetes Nuclear Transfer Embryonic Stem Cells. *Diabetes*.
- Yang, J. K., Lu, J., Yuan, S. S., Asan, Cao, X., Qiu, H. Y., Shi, T. T., Yang, F. Y., Li, Q., Liu, C. P., Wu, Q., Wang, Y. H., Huang, H. X., Kayoumu, A., Feng, J. P., Xie, R. R., Zhu, X. R., Liu, C., Yang, G. R., Zhang, M. R., Xie, C. L., Chen, C., Zhang, B., Liu, G., Zhang, X. Q. & Xu, A. 2018. From Hyper- to Hypoinsulinemia and Diabetes: Effect of KCNH6 on Insulin Secretion. *Cell Rep*, 25, 3800-3810.e6.

Reviewers' comments:

Reviewer #1 (Remarks to the Author):

My concerns have been addressed and I support publication.

Reviewer #2 (Remarks to the Author):

The authors state that they agree with my concerns and have addressed them but I find little evidence to support this in the revised manuscript.

The main weaknesses that I raised previously were:

- * The limits of the functional follow up on effects on insulin exocytosis
- * The fact several other manuscripts reporting the effects of HNF1A mutations/loss in stem cell models have been reported.
- * Lack of human data from carriers of HNF1A variants to support the observations that there is an altered ratio of Insulin/c-peptide.

The responses to these weaknesses has not really been addressed. They say they have removed reference to insulin exocytosis but the manuscript still talks about granule release rather than insulin secretion and in the absence of human data on the insulin/c-peptide phenotype the mouse findings require validation. Reviewer 1 has also highlighted some concerns here.

Furthermore, additional comments I raised have only superficially been addressed. For example

- * If the authors read the JCI paper they now cite they will read:-

"While 14 genes have now been designated as MODY genes in OMIM and/or the literature, three of these (BLK, PAX4, and KLF11) have been proposed for elimination based on a recent study (10) (see Table 1 for the remaining 11 along with RFX6, recently proposed as an additional MODY gene; ref. 11)."

Therefore the number of genes is not 14....

- * The authors still refer to sulphonyreas causing release of insulin from granules and state that sulphonlureas cause release of immature granules - something which they have not tested or shown.
- * In this reviewers opinion the effects on granule phenotype suggest a defect in maturation and would be expected to be accompanied by differences in proinsulin to insulin ratio.
- * The manuscript remains difficult to read and follow with lots of disjointed experiments.

Reviewer #3 (Remarks to the Author):

As already stated in the initial revision, the here presented study is overall very convincing and well performed. Although not all of the findings are of complete novelty, the data provides important insights on the molecular levels of the overall complex role of HNF1A in MODY. Overall the authors fulfilled the remarks of the initial revision. The incorporation of further publications of the field clearly improves the overall discussion sections. The figures are also more structured and unconvincing (FACS) data removed.

The following minor concerns should be addressed:

* In the figures the authors use the term "hESC HNF1A...". This is actually misleading, since you are referring to hESC-derived endocrine cells and not hESC

* Same also for Figure 2J, please note there that you stained grafts of the hESC-derived endocrine cells

* In Figure 6 the term MODY3 should also be replaced by the now used term HNF1A-MODY

* Some of the figure (figure 3, 4, 6) are not fully visible. They are truncated probably due to landscape format

"HNF1A deficiency causes reduced calcium levels and accumulation of abnormal insulin granules in a stem cell model of HNF1A-MODY"

COMMSBIO-21-1275A

Reviewers' comments:

Reviewer #1 (Remarks to the Author):

My concerns have been addressed and I support publication.

Response: We thank you for your efforts and patience; they have made this a better manuscript.

Reviewer #2 (Remarks to the Author):

The authors state that they agree with my concerns and have addressed them but I find little evidence to support this in the revised manuscript.

The main weaknesses that I raised previously were:

* The limits of the functional follow up on effects on insulin exocytosis

Response: We no longer focus on granule exocytosis as directly relevant to the results reported in the manuscript. When introducing the SYT13 gene in the results section, we mention the published literature on its role in calcium-mediated exocytosis.

* The fact several other manuscripts reporting the effects of HNF1A mutations/loss in stem cell models have been reported.

Response: We are of course well aware of these reports and acknowledge and comment on them in a dedicated paragraph in the Discussion. In the aggregate, our respective studies are complementary in that they point to convergent aspects of the molecular physiology of HNF1A in the beta cell. Our studies differ because they emphasize mechanistic implications with regard to calcium homeostasis. Importantly, our studies are unique in the demonstration of the effects of sulfonylureas – used clinically in the management of HNF1A-MODY – to elevate intracellular calcium in beta cells..

* Lack of human data from carriers of HNF1A variants to support the observations that there is an altered ratio of Insulin/c-peptide.

Response: We do have preliminary, retrospective data consistent with our inference based on the stem cell studies that the stoichiometry of insulin/c-peptide release would be altered in HNF1A-MODY patients. These results are, however, based on plasmas obtained in the context clinical management of patients in our diabetes center. We feel

that a prospective study of such subjects – ideally before and after pharmacological intervention (with e.g. sulfonylurea) is required before publication. Such studies would include formal glucose tolerance testing and, ideally, glucose clamping.

The responses to these weaknesses has not really been addressed. They say they have removed reference to insulin exocytosis but the manuscript still talks about granule release rather than insulin secretion and in the absence of human data on the insulin/c-peptide phenotype the mouse findings require validation. Reviewer 1 has also highlighted some concerns here.

Response: We no longer use the term exocytosis in the context of HNF1A phenotypes in the manuscript. We do present and discuss “accumulation of abnormal insulin granules” as observed in **Fig. 5A-5F** by electron microscopy.

We have made the following modification to the manuscript in the results section: “Thus, membrane depolarization by a sulfonylurea causes **secretion of insulin from HNF1A mutant β -cells that do not respond to glucose alone**. This distinction is consistent with the clinical efficacy of sulfonylurea drugs in HNF1A-MODY patients.”

The following sentence has been removed: “While these abnormal insulin granules are not secreted in response to glucose, insulin is secreted in response to sulfonylureas.”

Furthermore, additional comments I raised have only superficially been addressed. For example

* If the authors read the JCI paper they now cite they will read:-

“While 14 genes have now been designated as MODY genes in OMIM and/or the literature, three of these (BLK, PAX4, and KLF11) have been proposed for elimination based on a recent study (10) (see Table 1 for the remaining 11 along with RFX6, recently proposed as an additional MODY gene; ref. 11).”

Therefore the number of genes is not 14....

Response: The relevant text has been changed to: “There are at least 11 genes with mutations causing MODY.”

* The authors still refer to sulponlyreas causing release of insulin from granules and state that sulphonlureas cause release of immature granules - something which they have not tested or shown.

Response: We have amended the relevant texts as indicated above.

* In this reviewer's opinion the effects on granule phenotype suggest a defect in maturation and would be expected to be accompanied by differences in proinsulin to insulin ratio.

Response: This important point is addressed in the results section as follows: "The altered insulin:C-peptide ratio in grafted mice is not attributable to differences in insulin processing because insulin:proinsulin ratios in hESC *HNF1A* KO and hESC *HNF1A* WT grafts were identical *in vivo* (**Fig. 4E**) and *in vitro* (**Fig. S8J-S8L**), and no differences were found in the transcript levels of processing genes (*PC1/PC3*) in RNA sequencing analysis (**Table S3-S5**)."

All cells of all genotypes were analyzed after a minimum of 4 months of *in vivo* maturation as transplants, making it unlikely that immature cells (for which we found no evidence using immunocytochemistry of the grafts) can account for the differences in insulin:proinsulin ratios in plasma.

* The manuscript remains difficult to read and follow with lots of disjointed experiments.

Response: Figure S1 shows the progression of experiments and summarizes findings. We now point readers to this schematic throughout the manuscript. We also shortened the abstract.

Reviewer #3 (Remarks to the Author):

As already stated in the initial revision, the here presented study is overall very convincing and well performed. Although not all of the findings are of complete novelty, the data provides important insights on the molecular levels of the overall complex role of *HNF1A* in MODY.

Overall the authors fulfilled the remarks of the initial revision. The incorporation of further publications of the field clearly improves the overall discussion sections. The figures are also more structured and unconvincing (FACS) data removed.

The following minor concerns should be addressed:

* In the figures the authors use the term "hESC *HNF1A*...". This is actually misleading, since you are referring to hESC-derived endocrine cells and not hESC

Response: We have removed "hESC" from all figures and use only "HNF1A WT" or "HNF1A KO".

* Same also for Figure 2J, please note there that you stained grafts of the hESC-derived endocrine cells

Response: This point is now made explicit in the Figure and Legend

* In Figure 6 the term MODY3 should also be replaced by the now used term HNF1A-MODY

Response: Done

* Some of the figure (figure 3, 4, 6) are not fully visible. They are truncated probably due to landscape format

Response: This issue resulted from a processing error in manuscript conversion. The uploaded figures are not truncated.

REVIEWERS' COMMENTS:

Reviewer #2 (Remarks to the Author):

I thank the authors for taking another look at their response to my concerns and for addressing them consistently throughout the paper.